# Trading-off Statistical and Computational Efficiency via $W$-step MDPs: A Policy Gradient Approach

## Abstract

In reinforcement learning, algorithm performance is typically evaluated along two dimensions: computational and statistical complexity. While theoretical researchers often prioritize statistical efficiency—minimizing the number of samples needed to reach a desired accuracy—practitioners, in addition to the sample complexity, also focus on reducing computational costs, such as training time and resource consumption. Bridging these two perspectives requires algorithms able to deliver strong statistical guarantees while remaining computationally efficient in practice. In this paper, we introduce `MetaStep`, a meta-algorithm designed to enhance state-of-the-art RL algorithms by improving their computational efficiency while maintaining competitive sample efficiency. `MetaStep` is based on the novel notion of *W-step Markov decision process* (MDP), where, instead of performing a single action and transitioning to the next state, the agent executes a sequence of $W$ actions before observing the resulting state and collecting the discounted $W$-step cumulative reward. First, we provide a theoretical analysis of the suboptimality introduced in the optimal policy performance when planning in a $W$-step MDP, highlighting the impact of the environment stochasticity. Second, we apply `MetaStep` to GPOMDP, a well-known policy gradient method, and theoretically investigate the advantages of learning in the $W$-step MDP in terms of variance reduction and improved sample complexity. Finally, empirical evaluations confirm that `MetaStep` reduces computational costs while preserving—and, in certain scenarios, improving—sample efficiency.

## 1 Introduction

*Reinforcement learning* (RL, Sutton and Barto, 2018) has achieved remarkable success across a wide spectrum of complex real-world control tasks, such as robotic manipulation (Peters and Schaal, 2006), autonomous driving (Likmeta et al., 2020), game playing (Silver et al., 2018), and financial portfolio management (Jiang and Liang, 2017). Among RL methods, *policy gradient* (PG, Sutton et al., 1999a) approaches have gained significant popularity (Deisenroth et al., 2013) due to their simplicity (Silver et al., 2014), flexibility, and effectiveness (Peters and Schaal, 2008) in handling high-dimensional continuous control problems, which are prevalent in many real-world settings.

Despite these advances, implementing RL solutions in large-scale real-world control problems remains challenging due to two main limitations: sample inefficiency and high computational cost. Indeed, RL algorithms typically require extensive interaction with the environment to learn effective behaviors (Sutton and Barto, 2018), making them expensive and often impractical in resource-constrained scenarios. While theoretical research focuses on improving sample efficiency, practitioners often rely on simulated environments where data is abundant (Silver et al., 2016) and where the use of computationally inefficient learning algorithms to process the data may become the bottleneck.

One of the major challenges in RL is represented by the *curse of horizon* (Liu et al., 2018), which refers to the exponential growth in complexity (both statistical and computational) w.r.t. the decision horizon. As the time horizon extends, the estimation of future rewards becomes increasingly uncertain due to the accumulation of errors. This uncertainty can result in suboptimal behavior, as the impact of distant future rewards diminishes

compared to immediate rewards, complicating the learning process (Liu et al., 2020). This phenomenon impacts sample complexity, as longer horizons require more data to accurately estimate long-term returns and reduce variance in gradient-based policy search. From a computational complexity perspective, longer horizons demand more resources to simulate and backpropagate through lengthy decision sequences. This represents a serious challenge for time- or budget-constrained applications. To address the curse of horizon, several approaches have been explored from both perspectives. From a statistical complexity point of view, *Hierarchical RL* (Sutton et al., 1999b; Barto and Mahadevan, 2003; Pateria et al., 2022) methods create temporal abstractions in the action space, allowing for more structured decision-making and partitioning of the time horizon. Instead, *action-persistence* (Metelli et al., 2020; Sabbioni et al., 2023) focuses on identifying the best action to be repeated several times to reduce the effective planning horizon and consequently the sample complexity. However, forcing the same action to be repeated can sometimes oversimplify the problem and result in too much rigidity. Finally, *open-loop* approaches Bubeck and Munos (2010), which involve executing a sequence of actions without intermediate feedback, simplify the decision process over long horizons at the cost of poor performance in stochastic environments where reactive behavior is critical. Differently, from the computational complexity perspective, approaches like *frame skipping* (Kalyanakrishnan et al., 2021) and *action persistence* (Metelli et al., 2020), where an agent selects and repeats an action for several frames or time steps without further decisions, have been shown to reduce inference overhead by decreasing the number of decisions an agent must compute per episode. While effective in lowering computational demands, these approaches may compromise precision in tasks requiring fine-grained control.

In this paper, we tackle the curse of horizon problem in order to reduce both the *statistical* and *computational* complexity. Our goal is to bridge *action-persistence* and *open-loop* strategies to combine their complementary advantages: reducing the effective horizon while alleviating the computation burden during training. Action persistence offers various benefits, particularly in real-world control tasks where the optimal action evolves smoothly across states, as is common in many physical systems. On the other hand, open-loop strategies significantly reduce computational demands by eliminating the need for frequent state observations to influence the action selection process.

**Original Contribution.** This paper introduces the novel framework of $W$-*step Markov Decision Process* ($W$-MDP), where the agent plays sequences of $W$ actions without observing intermediate states. We design `MetaStep`, a meta-algorithm designed to be paired with other RL approaches, shifting the agent-environment interaction in $W$-MDPs. Eventually, we adopt `MetaStep` on top of GPOMDP (Baxter and Bartlett, 2001), presenting a statistically and computationally efficient variant to learn in this setting. The contributions of this work are summarized in more detail below.

- In Section 2, we present $W$-step Markov Decision Processes ($W$-MDPs). Then, we derive a novel result upper bounding the performance difference in terms of optimal value functions between planning in the original MDP and planning in the $W$-step MDP (Theorem 2.1), highlighting which features of the environment are responsible for the performance loss.
- In Section 3, we present `MetaStep`, our meta-algorithm, and we discuss how it translates into mutating the usual RL interaction protocol between agent and environment.
- In Section 4, we apply `MetaStep` on `GPOMDP`, a widely-adopted action-based policy gradient algorithm. We study and discuss its theoretical guarantees in terms of smoothness properties of the objective function and variance reduction, and show how these properties positively affect the convergence guarantees (Theorems 4.3 and 4.4).
- In Section 5, we numerically evaluate our solution over `GPOMDP` for different window sizes $W$ and discuss its empirical performance compared to the base implementation of `GPOMDP` both in terms of learning speed and computation time.

Related works are discussed in Section 6. The proofs of all the statements are provided in Appendix A. Additional experimental details are discussed in Appendix B.

## 2 Setting

In this section, we start by introducing the background notions on MDPs. Then, we present the $W$-MDP framework and the learning problem (Section 2.2). Finally, we propose a novel result on the optimal performance bias for $W$-MDPs (Section 2.3).

### 2.1 Preliminaries

**Markov Decision Processes.** A Markov Decision Process (MDP, Puterman, 1990) is defined as a tuple $\mathcal{M} := (\mathcal{S}, \mathcal{A}, P, r, \gamma, H, \mu_0)$, where $\mathcal{S}$ and $\mathcal{A}$ are the (continuous) state and action spaces, respectively, $P : \mathcal{S} \times \mathcal{A} \to \Delta(\mathcal{S})$ is the transition model, where $P(s'|s, a)$ specifies the probability of landing in state $s' \in \mathcal{S}$ after having played action $a \in \mathcal{A}$ in state $s \in \mathcal{S}$, $r : \mathcal{S} \times \mathcal{A} \to [-R_{\max}, R_{\max}]$ is the reward function, where $r(s, a)$ specifies the reward the agent gets by playing action $a$ in state $s$ and $R_{\max} \in \mathbb{R}_{\geq 0}$, $\gamma \in [0, 1]$ is the discount factor, $H \in \mathbb{N} \cup \{+\infty\}$ is the time horizon and $\mu_0 \in \Delta(\mathcal{S})$ is the initial-state distribution.[1] A trajectory $\tau = (s_0, a_0, ..., s_{H-1}, a_{H-1})$ of length $H$ is a sequence of $H$ state-action pairs. The *discounted return* of a trajectory $\tau$ is $R(\tau) := \sum_{h=0}^{H-1} \gamma^t r(s_h, a_h)$.

**Policies, Value Functions, and Optimality.** The agent's behavior is governed by a (possibly stochastic) Markovian policy $\pi : \mathcal{S} \to \Delta(\mathcal{A})$ where $\pi(a|s)$ is the probability to play action $a \in \mathcal{A}$ when in state $s \in \mathcal{S}$. The goal of an agent is to find the policy that maximizes the *expected cumulative discounted reward* $J(\pi) = \mathbb{E}_{\tau \sim d^\pi}[R(\tau)]$, where $d^\pi$ is the distribution over trajectories induced by policy $\pi$. Moreover, we define the *state value function* (Sutton and Barto, 2018) as the expected discounted cumulative reward induced by policy $\pi$ in state $s \in \mathcal{S}$: $V^\pi(s) = \mathbb{E}_\pi \left[ \sum_{h=0}^{H-1} \gamma^h r(s_h, a_h) \middle| s_0 = s \right]$. We define $\pi^*$ as an optimal policy as a policy maximizing the expected discounted reward for each state $s \in \mathcal{S}$: $\pi^* \in \operatorname{argmax}_{\pi \in \Pi} V^\pi(s)$, where $\Pi$ is the set of Markovian policies. We denote with $V^*(s) = V^{\pi^*}(s)$ the value function of an optimal policy $\pi^*$. We consider parametric policies $\pi_\theta$, i.e., policies characterized by a parameter vector $\theta$. We use the notation $J(\theta)$ to express the expected performance of policy $\pi_\theta$.

### 2.2 $W$-step Markov Decision Processes

Building on the definition of MDPs, we introduce the new concept of *$W$-step Markov decision processes* ($W$-MDPs), which redefines MDPs to handle $W$-step actions.

**Definition 2.1** ($W$-MDP). *Let $\mathcal{M}$ be an MDP. For any $W \in \mathbb{N}_{\geq 1}$, the $W$-MDP is defined as $\mathcal{M}_W = (\mathcal{S}, \mathcal{A}^W, P_W, r_W, \overline{H}, \overline{\gamma}, \mu_0)$, where $\mathcal{A}^W = \{\mathbf{a} = (a_1, \ldots, a_W) : a_i \in \mathcal{A} \ \forall i \in \{1, \ldots, W\}\}$ is the space of $W$-step actions, $P_W : \mathcal{S} \times \mathcal{A}^W \to \Delta(\mathcal{S})$ is the $W$-step transition model recursively defined as $P_1 = P$ and for $W \in \mathbb{N}_{\geq 2}$, $s \in \mathcal{S}$, and $(a_1, \ldots, a_W) \in \mathcal{A}^W$ as:*

$$P_W(\cdot|s, (a_1, \ldots, a_W)) = \int_{\mathcal{S}} P_{W-1}(s'|s, (a_1, \ldots, a_{W-1})) P(\cdot|s', a_W) \mathrm{d}s',$$

*$r_W : \mathcal{S} \times \mathcal{A}^W \to \left[ -\frac{1-\gamma^W}{1-\gamma} R_{max}, \frac{1-\gamma^W}{1-\gamma} R_{max} \right]$ is the $W$-step reward function defined as $r_1 = r$ and for $W \in \mathbb{N}_{\geq 2}$, $s \in \mathcal{S}$, and $(a_1, \ldots, a_W) \in \mathcal{A}^W$ as:*

$$r_W(s, (a_1, \ldots, a_W)) = r(s, a_1) + \gamma \int_{\mathcal{S}} P(s'|s, a_1) r_{W-1}(s', (a_2, \ldots, a_W)) \mathrm{d}s',$$

*$\overline{H} = \lceil H/W \rceil$ is the time horizon, and $\overline{\gamma} = \gamma^W$ is the discount factor.*

Therefore, this definition of $W$-MDP generalizes standard MDPs, as for $W = 1$ we recover the base MDP, i.e., $\mathcal{M} = \mathcal{M}_1$. In a $W$-step MDP, the policy is defined as function $\pi^W : \mathcal{S} \to \Delta(\mathcal{A}^W)$. At each decision step (i.e., when $h \bmod W = 0$), the agent samples a sequence of actions according to its policy and sequentially plays it for the next $W$ steps. The parameter $W \geq 1$ can be viewed as an environmental parameter (influencing $P$, $r$, $H$, and $\gamma$) that can be optimized to improve the agent's learning process. Notably, the effective horizon in $\mathcal{M}_W$ is reduced to $\frac{1}{1-\gamma^W} < \frac{1}{1-\gamma}$. The effect of open-loop actions is similar to reducing the planning horizon,

---

[1]We admit $H = \infty$ only for $\gamma < 1$.

either by explicitly reducing the task's discount factor and the trajectory length. As presented in Sections 4 and 5, the reduced effective horizon provably brings improvements both in terms of sample and computational complexity.

### 2.3 Performance bias on $W$-MDPs

The $W$-MDP allows for a lower effective horizon by aggregating steps. This aggregation is convenient in environments with deterministic transition probabilities but may weaken the performance when the environment is highly stochastic, as we operate partially open-loop, losing controllability. In this part, we study how much we lose in terms of the optimal value function because we are not fully able to control what happens in the intermediate steps and fully manage the environment's stochasticity.

Before that, we introduce the $W$-*step value function* $V_W^{\pi^W}$, which evaluates the performance of policy $\pi^W$ in the $W$-step MDP starting from any state $s \in \mathcal{S}$:

$$V_W^{\pi^W}(s) = \mathbb{E}_{\pi^W} \left[ \sum_{h=0}^{\overline{H}-1} \overline{\gamma}^h r_W(s_{Wh}, (a_{Wh}, \ldots, a_{Wh+W-1})) \middle| s_0 = s \right]. \tag{1}$$

An optimal policy $\pi^*$ is a policy maximizing the $W$-step value function from every state $s \in \mathcal{S}$, i.e., $\pi^* \in \operatorname{argmax}_{\pi^W \in \Pi^W} V_W^{\pi^W}(s)$, where $\Pi^W$ is the set of Markovian policies for the $W$-MDP. We define the *optimal $W$-value function* as $V_W^*(s) = V_W^{\pi^*}(s)$. We are now ready to bound the difference in the optimal value function when moving from the original MDP to the $W$-step MDP.

**Theorem 2.1.** *Let $\mathcal{M}$ be an MDP, let $W \in \mathbb{N}_{\geq 1}$, and let $\mathcal{M}_W$ be the corresponding $W$-step MDP. Then, for every $s \in \mathcal{S}$, it holds that:*

$$V^*(s) - V_W^*(s) \leq 2R_{max} \left( \frac{\gamma}{(1-\gamma)^2} - \frac{\gamma^W W}{(1-\gamma^W)^2} \right) D(P),$$

*where:*

$$D(P) := \max_{s,a \in \mathcal{S} \times \mathcal{A}} \left\{ \min_{s' \in \mathcal{S}} \left\{ 1 - P(s'|s,a) \right\} \right\}.$$

Some comments are in order. As expected, for $W = 1$, the performance difference between the $W$-step MDP and the normal MDP is equal to 0, i.e., $V^*(s) = V_1^*(s)$. Furthermore, we notice that the difference gap increases *linearly* with $D(P)$, which represents the index of stochasticity of the original transition model $P$. When $P$ is deterministic, then $D(P) = 0$ and, consequently, no suboptimality is introduced, i.e., $V^*(s) = V_W^*(s)$. This is compliant with the known fact that open-loop control is optimal for deterministic environments. Instead, when $P$ is "maximally stochastic", i.e., $P(s'|s,a) = 1/S$ for every $s' \in \mathcal{S}$, we have that $D(P) = 1 - 1/S$ taking its maximum value.

## 3 MetaStep

In this section, we present `MetaStep`, providing an overview of the meta-algorithm, how it interacts with the environment, and showing the differences w.r.t. classic agent-environment interaction.

The classic RL agent-environment interaction follows a continuous loop. Initially, the agent observes the current state $s$ of the environment. Based on this observation, the agent selects a (single-step) action $a$ following policy $\pi$ (i.e., $a \sim \pi(\cdot|s)$). Subsequently, the agent plays the action $a$, and the environment responds by transitioning to a new state $s'$ and providing the agent with feedback in the form of a reward $r$, reflecting the immediate benefit or cost of the action. This process repeats for a number of steps $H$, as the agent attempts to adapt its behavior based on the environment's responses.

Differently, in `MetaStep`, whose interaction protocol is presented in Algorithm 1, we redefine the communication protocol between the agent and the environment, as the agent interacts with an $W$-MDP (as defined in Definition 2.1). At each iteration, a sequence of $W \in \mathbb{N}_{\geq 1}$ actions is sampled from a policy $\pi^W$ observing only the state at time step $h$ when $h \bmod W = 0$. During the $W$ sequence of steps, we collect the cumulative discounted reward for $W$ steps and evaluate it as a single-step reward signal. Compared to the classic

---

**Algorithm 1:** `MetaStep` — Interaction Protocol.

---

**Input:** Horizon $H$, parameter $W$, policy $\pi^W$
**for** $h \in [\![\lceil H/W \rceil]\!]$ **do**
 Observe state $s$
 Sample action $\boldsymbol{a} = (a_1, \ldots, a_W) \sim \pi^W(\cdot|s)$
 **for** $w \in [\![W]\!]$ **do**
  | Play action $a_w$ and collect reward $r_w$
 **end**
**end**

---

interaction protocol, `MetaStep` uses an inner loop of $W$ iterations to manage the usage of actions and the collection of associated rewards, the next state $s'$ is observed only after $W$ steps and represents the consequence of the $W$ actions played in an open-loop sequence. This behavior implicitly brings a trade-off in our approach. From one side, we do not have to observe the state at each interaction, reducing the effective horizon of the task and lightening the computational burden during the training phase, as we will see in Section 5 (the effective horizon of the task is inversely proportional to the parameter $W$). On the other hand, without observing the current configuration of the environment, we could end up obtaining suboptimal performance, especially in highly stochastic environments, as detailed in Theorem 2.1.

## 4   `MetaStep` **on** `GPOMDP`

In this section, we investigate the performance of `MetaStep` when applied to `GPOMDP` (Baxter and Bartlett, 2001), which serves as a representative case due to its well-understood theoretical properties. We begin by illustrating how our meta-algorithm (Algorithm 1) can be instantiated within the `GPOMDP` framework, and then proceed to analyze its sample complexity.

**Preliminaries on Policy Gradient and `GPOMDP`.** Policy gradients are a class of RL algorithms for searching the best-performing policy over a class of parametrized differentiable stochastic policies $\Pi_\Theta = \{\pi_\theta : \mathcal{S} \to \Delta(\mathcal{A}) : \theta \in \Theta \subseteq \mathbb{R}^d\}$, where $\Theta$ is the parameter space. At each iteration $k \in [\![K]\!]$, a dataset of $N$ trajectories $\mathcal{T}_N^k = \{\tau_i\}_{i=1}^N$, where $N \in \mathbb{N}$ is the batch size, is collected using policy $\pi_{\theta_k}$. Then, the policy parameter is iteratively updated via gradient ascent:

$$\theta_{k+1} = \theta_k + \eta \widehat{\nabla}_\theta J(\theta_k), \tag{2}$$

where $\eta > 0$ is the step size and $\widehat{\nabla}_{\theta_k} J(\theta_k)$ is an estimate of the policy gradient using the batch $\mathcal{T}_N^k$. The most common policy gradient estimators are `REINFORCE` (Williams, 1992) and `GPOMDP` (Baxter and Bartlett, 2001)) that can be expressed as $\widehat{\nabla}_\theta J(\theta_k) = \frac{1}{N} \sum_{i=1}^N g(\tau_i|\theta_k)$ where $\tau_i \in \mathcal{T}_N^k$ for every $i \in [\![N]\!]$ and $g(\tau_i|\theta_k) = \nabla_\theta \log p(\tau_i|\theta_k) R(\tau_i)$, being $p(\cdot|\theta)$ the density function over trajetories induced by $\pi_\theta$ and $R(\tau)$ the trajectory return. In this work, $g$ will refer to the `GPOMDP` estimator, which is preferred over `REINFORCE` due to its smaller variance and is given by:

$$\widehat{\nabla}_\theta J(\theta) = \frac{1}{N} \sum_{i=1}^N \sum_{h=0}^{H-1} \gamma^t r(s_{i,h}, a_{i,h}) \sum_{u=0}^h \nabla_\theta \log \pi_\theta(a_{i,u}|s_{i,u}), \quad \tau_i \in \mathcal{T}_N^k, \;\; i \in [\![N]\!]. \tag{3}$$

Algorithm 2 reports the pseudocode of the `GPOMDP` in its base implementation.

### 4.1   **Running** `MetaStep` **on** `GPOMDP`

We now proceed to integrate `GPOMDP` (Algorithm 2) in the `MetaStep` procedure (Algorithm 1). The result is presented in Algorithm 3. Here, each episode is divided into $H/W$ steps. For each step, we observe the state $s_h$ and we sample a sequence of $W$ actions from the parametric $\pi_\theta^W$. Each action in the sequence is played, and the corresponding reward $r_w$ is observed. These rewards are aggregated into a so-called $W$-reward, which sums the rewards weighted by discount factors $\gamma^{w-1}$. The tuple $(s_h, \mathbf{a}_W, r_W)$ is then added to the trajectory $\tau_l$. After data collection, the algorithm estimates the gradient of the objective function $J(\theta)$ with respect

---

**Algorithm 2:** `GPOMDP`.

---

**Input :** Number of iterations $K$, batch size $N$, initial parameter vector $\theta_0$, environment $\mathcal{M}$, horizon $H$, discount factor $\gamma$, learning rate $\eta$

Initialize $\theta \leftarrow \theta_0$

**for** $i \in [\![K]\!]$ **do**

    Set the stochastic policy parameters: $\pi_\theta$

    **for** $l \in [\![N]\!]$ **do**

        Initialize trajectory $\tau_l$ as an empty tuple

        **for** $h \in [\![H]\!]$ **do**

            Observe state $s_h$

            Play action $a_h \sim \pi_\theta(\cdot|s_h)$ and observe reward $r_h$

            Add to $\tau_l$ the tuple $(s_h, a_h, r_h)$

        **end**

    **end**

    Estimate the gradient according to Equation (3)

    Update the policy parameter vector: $\theta \leftarrow \theta + \eta \widehat{\nabla}_\theta J(\theta)$

**end**

Return $\theta$

---

to $\theta$, using an estimator based on the observed trajectories and discounted rewards, following the standard `GPOMDP` estimator, applied to the $W$-MDP.

This new structure has several effects from both the computational and the statistical perspectives.

**Computational Complexity.** From a computational perspective, the proposed approach offers two key advantages. First, recalling the `GPOMDP` estimator function (Equation 3), adopting `MetaStep` makes the estimation faster as one of the summations is over $H/W$ terms instead of $H$. Second, we reduce the number of times we query the policy by a factor $W$. While the single queries may be more costly due to the extended temporal abstraction of the action, the overall reduction in query frequency leads to an improvement in the training phase, especially in settings where the policy evaluation might be a bottleneck. An extensive empirical evaluation of this phenomenon is provided in Section 5, where we examine the computational aspects across tasks of increasing complexity.

**Statistical Complexity.** Here, we analyze the statistical complexity of `MetaStep` over `GPOMDP`. The smaller discount factor $\gamma^W$ and the reduced effective horizon $H/W$ lower the variance caused by the long horizon, even if the single rewards are larger and the problem's *scale* does not change.

Let $K \in \mathbb{N}$ be the number of *iterations* and $N \in \mathbb{N}$ the batch size; given an accuracy threshold $\epsilon > 0$, our objective is to bound the *sample complexity* $NK$. We use $J^*(W)$ to refer to the optimal performance with parameter $W$, i.e., $J^*(W) = \sup_{\theta \in \Theta_W} J(\theta; W)$, and with $J(\theta_k; W)$ the performance obtained by the policy parametrization $\theta_k$ using parameter $W$.

To support the theoretical analysis, we present the following assumptions on the smoothness of the performance function, which are standard in the study of sample complexity (Yuan et al., 2022).

**Assumption 4.1** (Bounded Gradient). *There exist $G > 0$ such that $\forall W \in [\![H]\!]$ the expected norm squared of the gradient of $\log \pi_\theta^W(\cdot|s)$ satisfies:*

$$\mathbb{E}_{a \sim \pi_\theta^W(\cdot|s)} \left[ \left\| \nabla_\theta \log \pi_\theta^W(a|s) \right\|^2 \right] \leq G^2, \quad \forall s \in \mathcal{S}.$$

**Assumption 4.2** (Bounded Hessian). *There exist $F > 0$ such that $\forall W \in [\![H]\!]$ the expected spectral norm of the Hessian of $\log \pi_\theta^W(\cdot|s)$ satisfies:*

$$\mathbb{E}_{a \sim \pi_\theta^W(\cdot|s)} \left[ \left\| \nabla_\theta^2 \log \pi_\theta^W(a|s) \right\| \right] \leq F, \quad \forall s \in \mathcal{S}.$$

Intuitively, these assumptions ensure that the effect on the environment dynamics and reward (and on their gradient), after playing an action, is controllable. Given Assumptions 4.1 and 4.2, we can now state a bound

---

**Algorithm 3:** `MetaStep` on `GPOMDP`.

---

**Input :** Number of iterations $K$, batch size $N$, initial parameter vector $\theta_0$, parameter $W$, environment $\mathcal{M}_W$, horizon $H/W$, original discount factor $\gamma$, discount factor $\gamma^W$, learning rate $\eta$

Initialize $\theta \leftarrow \theta_0$

**for** $i \in [\![K]\!]$ **do**

    Set the stochastic policy parameters: $\pi_\theta$

    **for** $l \in [\![N]\!]$ **do**

        Initialize trajectory $\tau_l$ as an empty tuple

        **for** $h \in [\![H/W]\!]$ **do**

            Observe state $s_h$

            Sample action sequence $\mathbf{a}_W \sim \pi_\theta^W(\cdot|s_h)$

            **for** $w \in [\![W]\!]$ **do**

                Play action $a_w$ and observe reward $r_w$

                Update the $W$-reward $r_W \leftarrow r_W + \gamma^{w-1} r(s_{h+w}, a_w)$

            **end**

            Add to $\tau_l$ the tuple $(s_h, \mathbf{a}_W, r_W)$

        **end**

    **end**

    Compute the gradient estimator:

$$\widehat{\nabla}_\theta J(\theta) \leftarrow \frac{1}{N} \sum_{i=1}^{N} \sum_{h=0}^{\frac{H}{W}-1} \left( \sum_{u=0}^{h} \nabla_\theta \log \pi_\theta(\mathbf{a}_{\tau_i,u}|s_{\tau_i,u}) \right) \gamma^{Wh} r(s_{\tau_i,h}, \mathbf{a}_{\tau_i,h})$$

    Update the policy parameter vector: $\theta \leftarrow \theta + \eta \widehat{\nabla}_\theta J(\theta)$

**end**

Return $\theta$

---

on the variance of the gradient estimators and on the smoothness of the performance function for $W$-step MDPs.

**Lemma 4.1.** *Under Assumption 4.1, the variance of the `GPOMDP` estimator with batch size $N$ is bounded for every $W \in [\![H]\!]$ as:*

$$\mathbb{V}\mathrm{ar}[\widehat{\nabla}_\theta J(\theta; W)] \le \frac{R_{\max}^2}{(1-\gamma)^2} \frac{G^2(1-\gamma^H)}{N(1-\gamma^W)} \left[ 1 - \frac{H}{W} \left( \gamma^H - \gamma^{H+W} \right) - \gamma^H \right] := V.$$

**Lemma 4.2.** *Under Assumptions 4.1 and 4.2, $\forall W \in [\![H]\!]$, it holds that:*

$$\left\| \nabla^2 J(\theta) \right\| \le \frac{R_{max}}{(1-\gamma)} \frac{(G^2+F)}{1-\gamma^W} \left[ 1 - \frac{H}{W} \left( \gamma^H - \gamma^{H+W} \right) - \gamma^H \right] := L.$$

Lemmas 4.1 and 4.2 are needed in the analysis of sample complexity to guarantee a structure on the MDP (Yuan et al., 2022). We refer to Appendix A for the proof of Lemmas 4.1 and 4.2.

We highlight that the bound described in Lemma 4.1 does not grow as $W$, demonstrating how the usage of $W$-MDPs helps reduce the effective horizon of the task and consequently also the variance of the gradient estimates. In Figure 1, we show the *bias variance* trade-off as a function of $W$. We notice how the bias on the performance (stated in Theorem 2.1) follows a sublinearly increasing trend with $W$. Differently, the variance of the gradient estimation (Lemma 4.1) shows a descending trend as $W$ increases. These effects result in a U-shaped behavior, highlighting the trade-off between estimation accuracy and stability when we are in noisy environments. This highlights the importance of selecting an appropriate $W$, as for stochastic environments, it trades off bias and variance.

Below, we state the convergence on a stationary point.

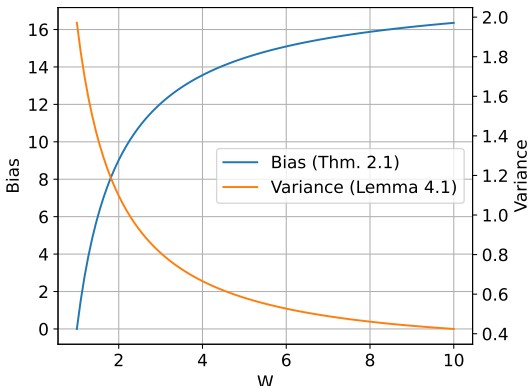

Figure 1: Trend of the variance of the gradient estimator as the parameter $W$ changes.

**Theorem 4.3.** *Consider an algorithm using the update rule of Eq.* (2). *Under Assumptions 4.1 and 4.2, with a suitable step size to guarantee* $\mathbb{E}\left[\|\nabla J(\theta; W)\|\right] \leq \epsilon$, *the sample complexity is at most:*

$$NK \leq \mathcal{O}\left(\frac{R_{\max}^3(1-\gamma^H)}{\epsilon^2(1-\gamma)^3(1-\gamma^W)^2}\left[1 - \frac{H}{W}\left(\gamma^H - \gamma^{H+W}\right) - \gamma^H\right]^2\right).$$

This result follows the analysis by Papini (2020). Here, the first term dominates, yielding the well-known $\Omega(\epsilon^{-2})$ rate. To also provide a global convergence result, we enforce the following assumption.

**Assumption 4.3** (Weak gradient domination for $J(W)$). *There exists $\alpha > 0$ and $\beta > 0$ such that for every $\theta \in \Theta_W$, and for every $W \in [\![H]\!]$, it holds that:*

$$J^*(W) - J(\theta; W) \leq \alpha \left\|\nabla_\theta J(\theta; W)\right\| + \beta.$$

Assumption 4.3 is a standard for global convergence analysis of stochastic optimization (Yuan et al., 2022). The weak gradient domination (WGD) assumption admits that the performance function $J(\theta_k; W)$ has local maxima as long as its performance is $\beta$-close to the globally optimal one. Given Assumption 4.3 and the results provided by Montenegro et al. (2024), we can now state the global convergence result.

**Theorem 4.4.** *Under Assumptions 4.1, 4.2 and 4.3, with a suitable step size to guarantee* $\mathbb{E}\left[J^*(W) - J(\theta_k; W)\right] \leq \epsilon + \beta$, *the sample complexity of Algorithm 3 is at most:*

$$NK \leq \widetilde{\mathcal{O}}\left(\frac{R_{\max}^3}{(1-\gamma)^3\epsilon^3}\frac{(1-\gamma^H)}{(1-\gamma^W)^2}\left[1 - \frac{H}{W}\left(\gamma^H - \gamma^{H+W}\right) - \gamma^H\right]^2\right).$$

This result establishes a convergence of order $\widetilde{\mathcal{O}}(\epsilon^{-3})$ to the last-iterate global optimum convergence of stochastic policies in accordance with state-of-the-art PG analysis (Yuan et al., 2022)[2]. We notice in the sample complexity how the usage of $W$-step MDPs manages to attenuate the curse of horizon by reducing the effective horizon by defining a new discount factor, $\gamma^W \geq \gamma$, which results in a lower sample complexity. The proofs of Theorems 4.3 and 4.4 are presented in Appendix A.

## 5   Experimental Validation

In this section, we show how `MetaStep` is effective in optimizing both the statistical and the computational complexity. We evaluate `MetaStep` applied to `GPOMDP` on three progressively complex domains to analyze its performance under different settings. For each environment, we test our approach with several values of $W \in \{1, 2, 3, 5, 10\}$, where $W = 1$ represents the standard version of `GPOMDP`. We adopt a stochastic Gaussian policy where the means for each action are parameterized using a neural network. Thus, we have a network

---

[2]The $\widetilde{\mathcal{O}}(\cdot)$ notation hides logarithmic factors.

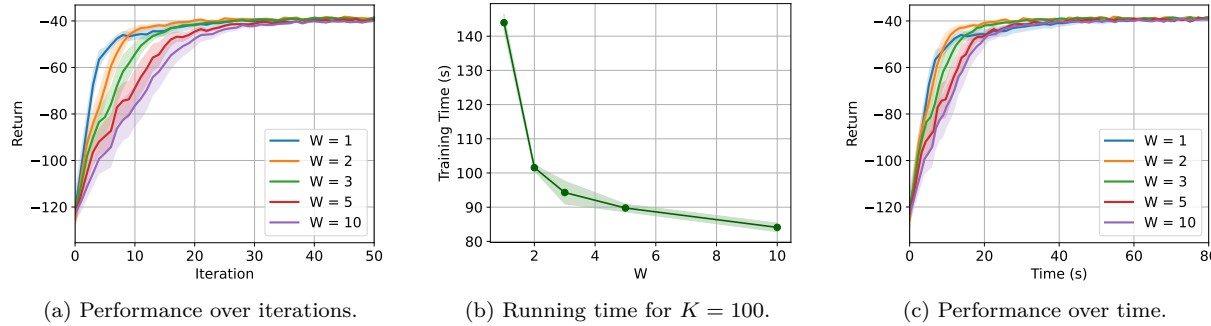

(a) Performance over iterations.        (b) Running time for $K = 100$.        (c) Performance over time.

Figure 2: Learning curves and training time for the dam control task (10 runs, mean $\pm$ 95% C.I.).

whose input is the state $s$ and the output is an action composed of $W d_{\mathcal{A}}$ elements. The standard deviation is fixed and tuned for each task.[3]

For each task, we will present three plots. The first (denoted with (a)) examines statistical complexity by illustrating the number of iterations (and thus the number of samples) required to achieve convergence for different values of $W$. The second plot (denoted with (b)) reports the training time as a function of the window size $W$. Lastly, the third plot (denoted with (c)) displays the learning curves relative to training time, offering insight into the computational complexity. We refer the reader to Appendix B for additional results and more detailed ablation study.

**Dam Control.** We first consider the water resource management task described in (Castelletti et al., 2010). The agent learns a water release policy balancing external demand $D$ (e.g., for a town) while preventing floods by keeping the water level below a threshold $F$. The dam experiences a stochastic daily inflow, representing external factors such as rainfall, which follow a periodic yearly pattern. The demand remains constant. The system's state evolves according to the mass balance equation $s_{t+1} = \max\{s_t - a_t + i_t, 0\}$ where $a_t$ is the water released on day $t$. The reward function, $R(s_t, a_t)$, is a convex combination of two objectives: flooding control, $-c_1 \max(0, s_t - F)$, and demand satisfaction, $-c_2 \max(0, D - a_t)^2$, with $c_1, c_2 > 0$ as domain-specific constants. The action space is scalar ($d_{\mathcal{A}} = 1$), while the state space has dimension $d_{\mathcal{S}} = 7$. The discount factor is $\gamma = 0.999$. We train our agent for $H = 1825$ steps (equivalent to 5 years of data) with batch size $N = 10$.

From the learning curves in Figure 2a, we observe that all approaches converge to the optimal value, regardless of the choice of $W$. Furthermore, Figure 2b shows how larger values of $W$ imply reduced training time, highlighting the computational advantages of our method. In the third view, presented in Figure 2c, we observe that the impact of wider windows over the training time is clear, and the improvement with $W \in \{2, 3\}$ is more evident w.r.t. the base algorithm ($W = 1$).

**Swimmer.** The Swimmer environment in MuJoCo (Todorov et al., 2012) is a continuous control problem where a multi-jointed agent must move itself forward in a simulated fluid environment. The action space dimension $d_{\mathcal{A}} = 2$, while the state space dimension $d_{\mathcal{S}} = 8$. We train our agent for $H = 700$ steps, using a batch size $N = 100$ and a discount factor $\gamma = 0.995$.

Figure 3a shows the learning curve over samples. We observe that larger values of $W$ lead to faster convergence and higher final performance. We conjecture that, this phenomenon is due to the search in the space of $\Pi^W$ policies, which induces a different manifold that might avoid plateaus during optimization. In Figure 3b, we appreciate the analysis of the computation time needed for the training as a function of $W$, highlighting its descending trend w.r.t. higher values of $W$. Finally, in Figure 3c, we present the performance over the training time required, highlighting the benefits of using a higher value of $W$.

---

[3]Appendix B provides additional experimental details and results. The code to reproduce the experiments is available in the supplementary material.

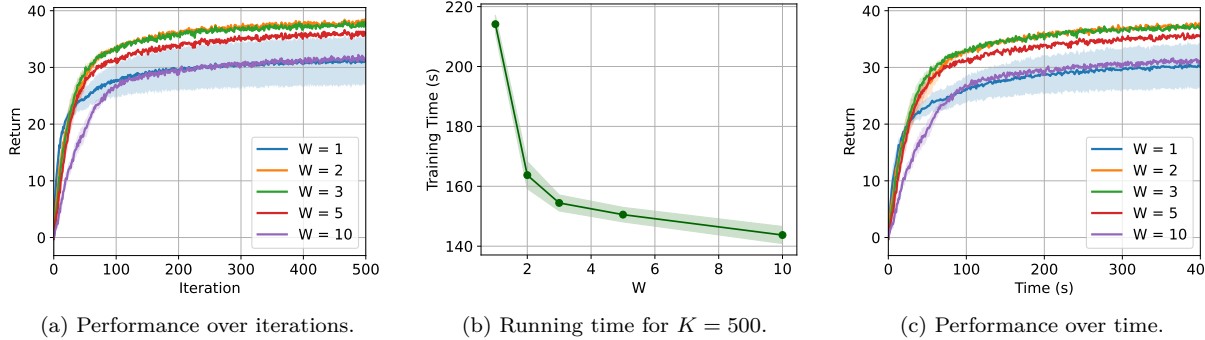

(a) Performance over iterations.        (b) Running time for $K = 500$.        (c) Performance over time.

Figure 3: Learning curves and training time for Swimmer (10 runs, mean $\pm$ 95% C.I.).

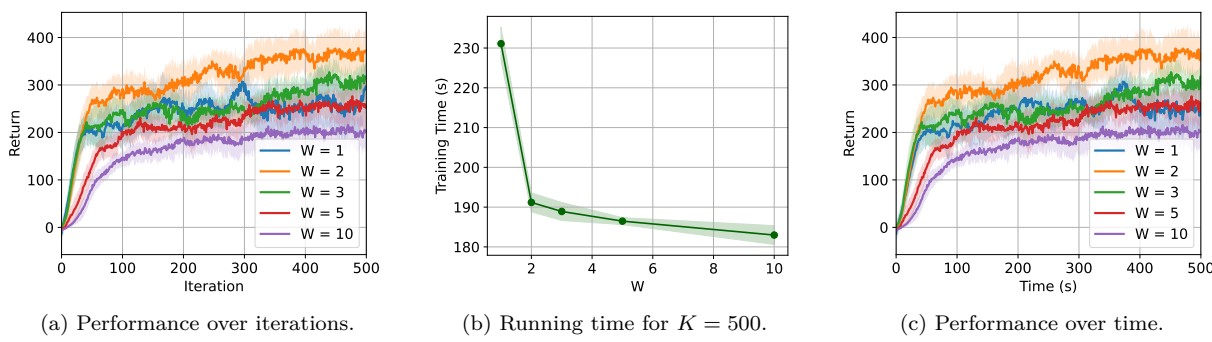

(a) Performance over iterations.        (b) Running time for $K = 500$.        (c) Performance over time.

Figure 4: Learning curves and training time for Half-Cheetah (10 runs, mean $\pm$ 95% C.I.).

**Half-Cheetah.** The Half-Cheetah environment in MuJoCo (Todorov et al., 2012) is a continuous control benchmark where a bipedal robot with a torso and two legs must learn to move forward efficiently. The action space has dimension $d_{\mathcal{A}} = 6$, while the state space has dimension $d_{\mathcal{S}} = 17$. We train the agent for $H = 200$ steps, using a batch size $N = 10$ and a discount factor $\gamma = 0.999$. The lower batch size, which leads to a noisy estimate of the gradients—especially in this high-dimensional scenario—is intended to evaluate the performance of `MetaStep` in this challenging situation.

In Figure 4a, we empirically validate Lemma 4.1, as with a smaller batch size, we manage to estimate a less noisy gradient and consequently reach better performances w.r.t. the single-step approach. Moreover, for farsighted problems (i.e., high values of $\gamma$), we enjoy a better sample complexity (as shown in Theorem 4.4). From the computational perspective, Figure 4b, also in this environment, demonstrates a clear reduction in training time as $W$ increases. Finally, in Figure 4c, we observe how, looking at the computational complexity, $W = 2$ presents a significantly faster convergence.

**Overall Results.** The experimental campaign demonstrates the effectiveness of the `MetaStep` approach in enhancing both statistical and computational efficiency. The results highlight a fundamental trade-off linked to the choice of the window parameter $W$. On the one hand, increasing $W$ reduces the problem's effective horizon. This leads to a decrease in the variance of the gradient estimate, as theorized in Lemma 4.1, and a significant reduction in training time. This often results in faster convergence, both in terms of wall-clock time and, in some cases, sample efficiency, as observed in the Swimmer and Half-Cheetah experiments. On the other hand, larger $W$ values increases the size of the action space, making the learning problem substantially more complex. The experiments show that while moderate values of $W$ (e.g., 2, 3) yield clear benefits, excessively high values (e.g., 10) can degrade final performance. This occurs because the difficulty of optimizing a policy in such a vast action space outweighs the advantages of a shorter horizon. The choice of $W$ represents a critical balance. An optimal value must shorten the horizon sufficiently to gain computational and stability benefits without expanding the action space to a point where learning becomes ineffective.

## 6 Related Works

In this section, we discuss the relevant literature for this work. First, we present closed-loop policy gradient approaches. Then, we discuss open-loop learning solutions and action persistence methods. Finally, we present temporal abstraction methods.

**Policy Gradient Methods.** Policy gradient (PG) methods constitute a fundamental class of reinforcement learning (RL) algorithms that directly optimize a parameterized policy by following the gradient of an expected performance measure. Early work by Williams (1992) introduced the REINFORCE algorithm, which laid the foundation for policy gradient techniques. Subsequent advancements, such as the GPOMDP algorithm (Baxter and Bartlett, 2001) and actor-critic approaches (Sutton et al., 1999a), addressed the high variance of gradient estimates by incorporating reward-to-go and baseline subtraction. In general, the problem of variance reduction in policy gradient estimation has attracted the attention of many researchers (Kakade, 2001; Schulman et al., 2015; Papini et al., 2018). Despite these advancements, computational efficiency remains challenging, as policy gradient methods typically require frequent updates and high sample complexity. Our work introduces a novel approach that mitigates these issues by leveraging open-loop sequences of actions, effectively reducing the number of required policy updates while maintaining competitive sample efficiency.

**Open-loop Approaches.** Open-loop planning in sequential decision problems refers to decision-making strategies where a sequence of actions is planned *entirely* in advance without incorporating immediate feedback from the environment at each step (Bubeck and Munos, 2010). This approach is particularly useful in domains where the real-time evaluation of actions is computationally expensive or in settings where the state of the environment is not (completely) observable (Yu et al., 2005). However, open-loop planning performs poorly in stochastic settings and non-generative models. To overcome this limitation, an approach of mixed planning has been proposed in literature (Hansen et al., 1996), mixing open-loop and closed-loop feedback planning. However, it provides only a value-based solution, which scales poorly with the dimensionality of the state and action space, with relevant limitations on the environment exploration. More recent solutions involve the mixed use of open-loop and closed-loop techniques constrained to specific operational regions of the task (Kolter et al., 2010). In our work, we propose a task-agnostic solution that embraces open-loop control methodologies but provides the agent with feedback after a fixed number of decision-making steps.

**Action Persistence.** Action persistence in sequential decision problems is a technique where an agent commits to executing the same action over multiple time steps. This approach has been shown to reduce the complexity of decision-making and improve learning efficiency in specific environments. Prior work has explored the effects of persistent actions in both value-based (Metelli et al., 2020; Sabbioni et al., 2023) and policy-based RL (Lee et al., 2020). More complex approaches have been proposed in the literature, which involve actor-critic structures (Yu et al., 2021) or Deep RL for policy approximation (Tong et al., 2023). However, these approaches tend to be too rigid and require properly setting the task control frequency. Furthermore, they poorly scale with the increase in dimensionality of the action space. Our work aligns with the philosophy of action persistence but provides more flexibility to the framework.

**Temporal Abstraction.** Temporal abstraction in RL aims to improve sample efficiency and planning capabilities by structuring decision-making at multiple time scales. Options (Sutton et al., 1999b) and macro-actions (McGovern and Barto, 2001) provide a hierarchical framework in which temporally extended actions facilitate long-term reasoning and efficient exploration. The MAXQ (Dietterich, 2000) is a classical example of hierarchical solution. At its core, MAXQ proposes a value function decomposition, dividing the main MDP into smaller sub-MDP components. In this way, each sub-MDP is associated with a sub-task that can be learned independently from the others. On the Deep RL side, (Kulkarni et al., 2016) proposed a solution consisting of a two-level hierarchy of Deep Q-Networks (DQN, Mnih et al., 2015) that represent the high and lower level policies, allowing for a hierarchical approach in large and high-dimensional state spaces. Another class of HRL approach involves both the learning of the hierarchical policy and the semi-MDP structure (*sub-task discovery*). The skill chaining approach (Konidaris and Barto, 2009), incrementally construct options while learning an HRL agent. This process starts with one option created with its own subgoal set. The usage of a classifier is exploited to define new options based on the possibility of reaching a

state in a predefined number of steps. More recently, meta-learning and learned subroutines (Bacon et al., 2017; Harb et al., 2018) have been employed to autonomously discover useful action sequences, reducing the complexity of long-horizon tasks. Specifically, Option Critic (OC, Bacon et al., 2017), let the agent use the sub-tasks (options) from the initial episode of learning. The solution proposed by OC allows to learn the options along with the entire hierarchy structure from the beginning of the learning process without the necessity of defining the hierarchical problem beforehand. In (Harb et al., 2018), a refinement of the OC, Asynchronous Advantage Option Critic (A2OC), has been proposed, introducing a regularizer cost that penalizes the high-level policy upon switching options, encouraging the retention of each option for a longer horizon. This approach proved empirically to have better performance than the frequently switching options. Other works, like (Barreto et al., 2019) adopts pseudo-rewards (e.g., cumulants) to represent option which can be combined linearly to synthesize new options without requiring additional learning.

While these methods effectively address long-term credit assignment, they often require significant computational resources and complex architectures. Our work aligns with temporal abstraction by introducing $W$-step action sequences that effectively function as macro-actions. However, unlike standard hierarchical RL methods, our approach does not require hierarchical structures or additional meta-learning components, making it a more lightweight and easily integrable enhancement to existing state-of-the-art algorithms. For a complete discussion on temporal abstraction, we refer the reader to (Pateria et al., 2022).

## 7    Discussion and Conclusions

In this paper, we introduced the novel concept of $W$-step Markov Decision Processes, which mixes open-loop and closed-loop interaction, and designed `MetaStep` a meta-algorithm exploiting the $W$-MDPs framework. In this framework, we observe a state and execute a sequence of $W$ actions in an open-loop fashion. We formalized the $W$-MDP framework and we define the notions of value function and optimality. Then, we showed how we may experience a suboptimality bias for highly stochastic environments. Building on this, we proposed a *meta-algorithm*, `MetaStep`, which redefines the agent-environment interaction by applying it over $W$-MDPs. `MetaStep` is designed to be used on top of standard RL approaches, and, in this paper, we adopted `GPOMDP` as a base learning algorithm. We discussed the improvement of the solution from both the computational and the statistical perspectives. Finally, we proposed an extensive experimental evaluation to empirically validate our results. Such results demonstrate the capability of our approach to keep competitive statistical performances while reducing the computational complexity.

**Limitations and Future Works.**    While the use of the $W$-MDPs framework shows promising results both theoretically (Sections 2 and 4) and empirically (Section 5), the approach still has some limitations. First, it currently supports only fixed values of $W$, leaving the choice of this hyperparameter to human expertise. The window size must balance several factors, such as the bias–variance trade-off (Figure 1), which is particularly important in highly stochastic environments (or where the level of stochasticity is unknown), and the trade-off between a shorter effective horizon and an expanded action space. Future works may explore adaptive window sizes, as relying on a fixed open-loop window can be overly restrictive in tasks requiring finer control. Another direction is to study the applicability of `MetaStep` to other state-of-the-art policy search methods, including actor–critic approaches. Finally, extending the framework to large-scale or real-world domains (e.g., robotics or control of complex systems) would offer valuable insights into its practical impact.

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

## A Omitted Proofs

In this appendix, we provide the proofs of all the theorems presented in the main paper.

**Theorem 2.1.** *Let $\mathcal{M}$ be an MDP, let $W \in \mathbb{N}_{\geq 1}$, and let $\mathcal{M}_W$ be the corresponding $W$-step MDP. Then, for every $s \in \mathcal{S}$, it holds that:*

$$V^*(s) - V_W^*(s) \leq 2R_{max} \left( \frac{\gamma}{(1-\gamma)^2} - \frac{\gamma^W W}{(1-\gamma^W)^2} \right) D(P),$$

*where:*

$$D(P) := \max_{s,a \in \mathcal{S} \times \mathcal{A}} \left\{ \min_{s' \in \mathcal{S}} \left\{ 1 - P(s'|s,a) \right\} \right\}.$$

*Proof.* Let us first introduce *state-action value function* as the expected discounted cumulative reward in state $s \in \mathcal{S}$ when we chose action $a \in \mathcal{A}$ and then follow policy $\pi$:

$$Q^\pi(s,a) = \mathbb{E}_\pi \left[ \sum_{h=0}^{H-1} \gamma^h r(s_h, a_h) \bigg| s_0 = s, a_0 = a \right]. \tag{4}$$

For a $W$-MDP, the definition of $Q_W^{\pi^W}$ follows directly.

We start the proof by observing that, for every MDP and corresponding $W$-MDP, we have $Q_1^*(s,a) \geq Q_W^*(s,a)$.

Then, we consider the non-stationary transition model defined as follows for every $h$:

$$\overline{P}_W(s'|s,a,h) \begin{cases} P(s'|s,a) & \text{if } h \bmod W = 0 \\ \mathbb{1}\left\{\text{argmax}_{\bar{s} \in \mathcal{S}} P(\bar{s}|s,a)\right\} & \text{otherwise} \end{cases}$$

We now highlight the dependence on the $W$ and on the used transition model $\overline{P}$ in the $Q$ as $Q_{W,\overline{P}}^*(s,a)$. It is trivial to observe that for this non-stationary deterministic value function, we have:

$$Q_{1,\overline{P}_W}^*(s,a) = Q_{W,\overline{P}_W}^*(s,a),$$

as where we are not able to take action, we have deterministic transitions. This is equal to say that in the MDP with $\overline{P}_W$ we have:

$$\max_{\pi:\mathcal{S} \to \mathcal{A}} \mathbb{E} \left[ \sum_h \gamma^h r_h \right] = \max_{\pi^W:\mathcal{S} \to \mathcal{A}^W} \mathbb{E} \left[ \sum_h \gamma^h r_h \right].$$

Now the goal is to try to bound the difference between $Q_{1,P}^*$ and $Q_{W,P}^*$. This is equal to:

$$\begin{aligned} Q_{1,P}^* - Q_{W,P}^* &= Q_{1,P}^* - Q_{W,P}^* \pm Q_{W,\overline{P}_W}^* \\ &\leq Q_{1,P}^* - Q_{1,\overline{P}_W}^* + Q_{W,\overline{P}_W}^* - Q_{W,P}^* \\ &\leq Q_{1,P}^{\pi^{1*}} - Q_{1,\overline{P}_W}^{\pi^{1*}} + Q_{W,\overline{P}_W}^{\overline{\pi}^{W*}} - Q_{W,P}^{\overline{\pi}^{W*}} \end{aligned}$$

Since we are to bound the difference between two Q-functions of the same policy under different transition models, we proceed by analyzing the first term only, as the second one is analogous:

$$Q_1^{\pi^*,P} - Q_1^{\pi^*,\overline{P}_W} = \mathbb{E}_{\bar{P}} \left[ \sum_{t=0}^\infty \bar{\gamma}^t r(s_t, a_t) \right] - \mathbb{E}_P \left[ \sum_{t=0}^\infty \gamma^t r(s_t, a_t) \right] \tag{5}$$

$$= \int_\tau \left( \prod_{l=0}^\infty P(s_{l+1}|s_l, \underline{a}_l) \pi^{1*}(a_l|s_l) - \prod_{l=0}^\infty \overline{P}_W(s_{l+1}|s_l, a_l, t) \pi^{1*}(a_l|s_l) \right) \sum_{t=0}^\infty \gamma^t r(s_t, a_t) d\tau \tag{6}$$

$$= \sum_{t=0}^\infty \gamma^t \int_\tau \left( \prod_{l=0}^t P(s_{l+1}|s_l, a_l) \pi^{1*}(a_l|s_l) - \prod_{l=0}^t \overline{P}_W(s_{l+1}|s_l, a_l) \pi^{1*}(a_l|s_l) \right) r(s_t, a_t) d\tau \tag{7}$$

$$= \sum_{t=0}^{\infty} \gamma^t \int_{\tau} \sum_{v=0}^{t-1} \prod_{l=0}^{v-1} P(s_{l+1}|s_l,a_l)\pi^{1*}(a_l|s_l)(P(s_{v+1}|s_v,a_v,v) - \overline{P}_W(s_{v+1}|s_v,\underline{a}_v))\pi^{1*}(a_v|s_v) \tag{8}$$

$$\cdot \prod_{l=v+1}^{H} \overline{P}_W(s'_{l+1}|s_l,a_l)\pi^{1*}(a_l|s_l)r(s_t,a_t)d\tau \tag{9}$$

$$= R_{\max} \sum_{t=0}^{\infty} \gamma^t \sum_{v=0}^{t-1} D_v = (*) \tag{10}$$

where:

$$D_v := \max_{s,a} \left\| P(\cdot|s,a,v) - \overline{P}_W(\cdot|s,a) \right\|_1 \tag{11}$$

$$= \max_{s,a} \int_{s'} (\bar{P}(\cdot|s,a,v) - \overline{P}_W(\cdot|s,a))ds' \tag{12}$$

$$\leq D\mathbb{1}\{v \mod W \neq 0\}. \tag{13}$$

Eventually, we have:

$$(*) = DR_{\max} \sum_{t=0}^{\infty} \gamma^t \sum_{v=0}^{t-1} \mathbf{1}\{v \mod W \neq 0\} \tag{14}$$

$$= DR_{\max} \left( \frac{\gamma}{(1-\gamma)^2} - \frac{\gamma^W W}{(1-\gamma^W)^2} \right). \tag{15}$$

$\square$

**Lemma 4.1.** *Under Assumption 4.1, the variance of the* `GPOMDP` *estimator with batch size N is bounded for every* $W \in [\![H]\!]$ *as:*

$$\mathbb{V}\mathrm{ar}[\widehat{\nabla}_\theta J(\theta;W)] \leq \frac{R_{\max}^2}{(1-\gamma)^2} \frac{G^2(1-\gamma^H)}{N(1-\gamma^W)} \left[ 1 - \frac{H}{W}\left(\gamma^H - \gamma^{H+W}\right) - \gamma^H \right] := V.$$

*Proof.* Starting from Lemma A.1, recalling the Assumption A.2 and A.3, we transpose the lemma in $W$-step MDPs, applying Definition 2.1 and Assumption 4.1, thus $H \leftarrow \frac{H}{W}$, $\gamma \leftarrow \gamma^W$ and $R_{\max} \leftarrow R_{\max,W}$, as

$$\mathbb{V}\mathrm{ar}[\widehat{\nabla}_\theta J(\theta)] \leq \frac{R_{\max}^2}{(1-\gamma)^2} \frac{G^2(1-\gamma^H)}{N(1-\gamma^W)} \left[ 1 - \frac{H}{W}\left(\gamma^H - \gamma^{H+W}\right) - \gamma^H \right]$$

$\square$

**Lemma 4.2.** *Under Assumptions 4.1 and 4.2,* $\forall W \in [\![H]\!]$, *it holds that:*

$$\left\| \nabla^2 J(\theta) \right\| \leq \frac{R_{max}}{(1-\gamma)} \frac{(G^2+F)}{1-\gamma^W} \left[ 1 - \frac{H}{W}\left(\gamma^H - \gamma^{H+W}\right) - \gamma^H \right] := L.$$

*Proof.* Recalling Assumption 4.1 and 4.2 and given Lemma A.2 applying Definition 2.1, thus $H \leftarrow \frac{H}{W}$, $\gamma \leftarrow \gamma^W$ and $R_{\max} \leftarrow R_{\max,W}$, we get,

$$\left\| \nabla^2 J(\theta) \right\| \leq \frac{R_{\max}}{(1-\gamma)} \frac{(G^2+F)}{1-\gamma^W} \left[ 1 - \frac{H}{W}\left(\gamma^H - \gamma^{H+W}\right) - \gamma^H \right]$$

$\square$

**Theorem 4.3.** *Consider an algorithm using the update rule of Eq. (2). Under Assumptions 4.1 and 4.2, with a suitable step size to guarantee* $\mathbb{E}\left[\|\nabla J(\theta;W)\|\right] \leq \epsilon$, *the sample complexity is at most:*

$$NK \leq \mathcal{O}\left( \frac{R_{\max}^3(1-\gamma^H)}{\epsilon^2(1-\gamma)^3(1-\gamma^W)^2} \left[ 1 - \frac{H}{W}\left(\gamma^H - \gamma^{H+W}\right) - \gamma^H \right]^2 \right).$$

*Proof.* We first apply Theorem A.3 under Assumptions A.1, A.2 and A.3.

$$NK = 2\left(J^*(W) - J(\theta_0; W)\right)\left(\frac{ZN}{\epsilon} + \frac{Z\nu_1}{\epsilon^2}\right)$$

We redefine $V \leftarrow \nu_1$ and $L \leftarrow Z$, applying Lemma 4.2 and 4.1. Then, neglecting constants, we obtain

$$NK = \frac{R_{\max}N}{\epsilon(1-\gamma)(1-\gamma^W)}\left[1 - \frac{H}{W}\left(\gamma^H - \gamma^{H+W}\right) - \gamma^H\right]$$
$$+ \frac{R_{\max}^3(1-\gamma^H)}{\epsilon^2(1-\gamma)^3(1-\gamma^W)^2}\left[1 - \frac{H}{W}\left(\gamma^H - \gamma^{H+W}\right) - \gamma^H\right]^2$$

□

**Theorem 4.4.** *Under Assumptions 4.1, 4.2 and 4.3, with a suitable step size to guarantee* $\mathbb{E}\left[J^*(W) - J(\theta_k; W)\right] \leq \epsilon + \beta$, *the sample complexity of Algorithm 3 is at most:*

$$NK \leq \widetilde{\mathcal{O}}\left(\frac{R_{\max}^3}{(1-\gamma)^3\epsilon^3}\frac{(1-\gamma^H)}{(1-\gamma^W)^2}\left[1 - \frac{H}{W}\left(\gamma^H - \gamma^{H+W}\right) - \gamma^H\right]^2\right).$$

*Proof.* We first apply Theorem A.4, under Assumptions A.5, A.6 and A.7.

$$NK = \frac{16\alpha^4 L_2 V_A}{\epsilon^3}\log\frac{\max\{0, J^*(W) - J(\theta_k) - \beta\}}{\epsilon}$$

We redefine $V \leftarrow V_A$ and $L \leftarrow L_{2,A}$, applying Lemma 4.2 and 4.1. Then, neglecting logarithmic values and constants, we obtain

$$NK = \frac{R_{\max}^3}{(1-\gamma)^3}\frac{(1-\gamma^H)}{\epsilon^3(1-\gamma^W)^2}\left[1 - \frac{H}{W}\left(\gamma^H - \gamma^{H+W}\right) - \gamma^H\right]^2$$

□

## A.1 Technical Lemmas and Theorems used in the Analysis

In this part, we report assumptions, theorems, and lemmas from other works we used for the analysis.

**Assumption A.1** (Papini et al. 2022, Definition 1)**.** *Let* $\Pi_\Theta = \{\pi_\theta \mid \theta \in \Theta\}$ *be a class of twice-differentiable parametric stochastic policies, where* $\Theta \subset \mathbb{R}^d$ *is convex. There exist a constant* $\xi_1 > 0$ *such that for every state* $s \in S$ *is defined as,*

$$\mathbb{E}_{a\sim\pi_\theta(\cdot|s)}\left[\left\|\nabla_\theta\log\pi_\theta(a|s)\right\|\right] \leq \xi_1,$$

**Assumption A.2** (Papini et al. 2022, Definition 1)**.** *Let* $\Pi_\Theta = \{\pi_\theta \mid \theta \in \Theta\}$ *be a class of twice-differentiable parametric stochastic policies, where* $\Theta \subset \mathbb{R}^d$ *is convex. There exist a constant* $\xi_2 > 0$ *such that for every state* $s \in S$ *is defined as,*

$$\mathbb{E}_{a\sim\pi_\theta(\cdot|s)}\left[\left\|\nabla_\theta\log\pi_\theta(a|s)\right\|^2\right] \leq \xi_2, \tag{16}$$

**Assumption A.3** (Papini et al. 2022, Definition 1)**.** *Let* $\Pi_\Theta = \{\pi_\theta \mid \theta \in \Theta\}$ *be a class of twice-differentiable parametric stochastic policies, where* $\Theta \subset \mathbb{R}^d$ *is convex. There exist a constant* $\xi_3 > 0$ *such that for every state* $s \in S$ *is defined as,*

$$\mathbb{E}_{a\sim\pi_\theta(\cdot|s)}\left[\left\|\nabla_\theta^2\log\pi_\theta(a|s)\right\|\right] \leq \xi_3, \tag{17}$$

**Assumption A.4** (Yuan et al. 2022, Assumption 3.1)**.** *There exists* $L > 0$ *such that, for all* $\theta, \theta' \in \mathbb{R}^d$, *it holds:*

$$\left|J(\theta') - J(\theta) - \langle\nabla J(\theta), \theta' - \theta\rangle\right| \leq \frac{I}{2}\left\|\theta' - \theta\right\|^2.$$

**Assumption A.5** (Montenegro et al. 2024, Assumption 6.1)**.** *There exist* $\alpha > 0$ *and* $\beta \geq 0$ *such that for every* $\theta \in \Theta$ *it holds that*

$$J_\theta^* - J(\theta) \leq \alpha\|\nabla_\theta J(\theta)\| + \beta.$$

**Assumption A.6** (Montenegro et al. 2024, Assumption 6.2). *$J_\theta$ is $L_{2,\mu}$-Lipschitz smooth w.r.t. parameters $\theta$, i.e., for every $\theta, \theta' \in \Theta$:*

$$\|\nabla_\theta J(\theta) - \nabla_\theta J(\theta')\| \le L_{2,\mu}\|\theta - \theta'\|.$$

**Assumption A.7** (Montenegro et al. 2024, Assumption 6.3). *For every $\theta \in \Theta$, the stochastic gradient $\widehat{\nabla}_\theta J(\theta)$ computed with batch size $N$ has bounded variance, i.e.,*

$$\mathbb{E}\big[\|\widehat{\nabla}_\theta J(\theta) - \nabla_\theta J(\theta)\|^2\big] \le \frac{V}{N}.$$

**Lemma A.1** (Papini et al. 2022, Lemma 29). *Given Assumptions A.2 and A.3 and a task horizon $H$. for every $\theta \in \Theta$, the variance of the* GPOMDP *estimator is upper-bounded as follows:*

$$\mathbb{Var}[\widehat{\nabla}_\theta J(\theta)] \le \frac{\xi_2 R_{max}^2 (1-\gamma^H)}{N(1-\gamma)^3} \big[1 - H(\gamma^H - \gamma^{H+1}) - \gamma^H\big] := \nu$$

*Proof.* The result follows the one of (Papini et al., 2022) Lemma 29, stopping one step before on line D82. □

**Lemma A.2** (Yuan et al. 2022, Lemma 4.4). *Under Assumptions A.2 and A.3 $J(\cdot)$ is $L$-smooth, namely $\big\|\nabla^2 J(\theta)\big\| \le L$ for all $\theta$ which is a sufficient of Assumption A.4 with*

$$I = \frac{R_{max}}{(1-\gamma^2)}(\xi_2 + \xi_3)$$

**Theorem A.3** (Papini 2020, Theorem 7.1). *Under Assumptions A.1, A.2 and A.3, running an algorithm with the update rule of Eq. (3) with initial policy parameters $\theta_0$, batch size $N$, and a step size $\eta = min\big\{\frac{1}{Z}, \frac{\epsilon N}{ZV}\big\}$*

$$NK = 2(J(\theta^*) - J(\theta_0))\left(\frac{Z}{\epsilon} + \frac{Z\nu}{\epsilon^2 N}\right)$$

*guarantees $\mathbb{E}[\|\nabla_\theta J(\theta)\|] \le \epsilon$ for a $k$ uniformly sampled from $\{0,1,...,K-1\}$, where $Z = \frac{R_{max}}{(1-\gamma)^2}\left(\frac{2\gamma\xi_1^2}{1-\gamma} + \xi_2 + \xi_3\right)$ and $\nu$ follows the definition of Lemma A.1.*

**Theorem A.4** (Montenegro et al. 2024, Theorem F.1). *Under Assumptions A.5, A.6 and A.7, running Algorithm 2 for $K > 0$ iterations with a batch size of $N > 0$ trajectories in each iteration with the constant learning rate $\eta$ fulfilling:*

$$\eta \le \min\left\{\frac{1}{L_2}, \frac{1}{\mu \max\{0, J^* - J(\theta_0) - \beta\}}, \left(\frac{N}{L_2 V_A \mu}\right)^{1/3}\right\}$$

*where $\mu = \frac{1}{\alpha^2}$. Then, it holds that:*

$$J^* - \mathbb{E}[J(\theta_K)] \le \beta + \left(1 - \frac{1}{2}\sqrt{\frac{\mu\eta^3 L_2 V_A}{N}}\right)^K \max\{0, J^* - J(\theta_0) - \beta\} + \sqrt{\frac{L_2 V_A \eta}{\mu N}}.$$

*In particular, for sufficiently small $\epsilon > 0$, setting $\eta = \frac{\epsilon^2 \mu N}{4L_2 V}$, the following total number of samples is sufficient to ensure that $J(\theta^*) - \mathbb{E}[J(\theta_K)] \le \beta + \epsilon$*

$$NK \ge \frac{16L_2 V_A}{\epsilon^3 \mu^2} log\frac{max\{0, J^* - J(\theta_0) - \beta\}}{\epsilon}$$

We refer to (Montenegro et al., 2024) for the formal definition of $L_2$ and $V_A$.

# B Experimental Settings and Additional Results

In this appendix, we discuss the experimental setting for the simulations provided in the main paper, and we provide further results.

## B.1 Experimental setting

**Algorithm Settings.** To further reduce the variance in estimating the gradients, we adopt a version of GPOMDP with the optimal baseline (Peters and Schaal, 2008):

$$
\hat{\nabla}_{\theta_t} J_{\theta_t} = \mathbb{E}_{p_\theta(\tau)} \left[ \sum_{j=0}^{T-1} \sum_{t=0}^{j} \nabla_\theta \log \pi_\theta \left( a_t \mid s_t \right) \left( r_j - b_j \right) \right],
$$
$$
b_j = \frac{\mathbb{E}_{p_\theta}(\tau) \left[ \left( \sum_{t=0}^{j} \nabla_{\theta_h} \log \pi_\theta \left( a_t \mid s_t \right) \right)^2 r_j \right]}{\mathbb{E}_{p_\theta}(\tau) \left[ \left( \sum_{t=0}^{j} \nabla_{\theta_h} \log \pi_\theta \left( a_t \mid s_t \right) \right)^2 \right]}. \tag{18}
$$

The policy adopted for the experiments has been parametrized using a neural network with 2 hidden layers, composed of 50 and 25 parameters each.

We tested our algorithm on 3 environments of increasing complexity. *Dam control* (Castelletti et al., 2010); *Swimmer-v4*, *HalfCheetah-v4* from the MuJoCo suite (Todorov et al., 2012). Details on the environmental parameters are shown in Table 1. We adopted different learning rates $\eta$ and exploration parameters, depending on the task we were observing, employing in some cases Adam (Kingma and Ba, 2015) to adaptively set the step size. The exact method adopted for each environment is shown in Table 2.

**Computational Resources.** All the experiments are run on a MacBook Pro. The machine is equipped as follows:

- CPU: Apple M2 Pro (10 cores, 3.4 GHz);

- RAM: 16 GB;

- GPU: 16-core GPU.

All the performances are run over 8 CPU cores. Refer to Figures 2b, 3b and 4b for the computational times for each environment for relevant experiments.

| Environment | Epoch | $N$ | $H$ | $\gamma$ | $d_{\mathcal{S}}$ | $d_{\mathcal{A}}$ |
|---|---|---|---|---|---|---|
| Dam control | 100 | 10 | 1825 | 0.999 | 7 | 1 |
| Swimmer | 500 | 10, 100 | 200, 500, 700 | 0.995, 0.999 | 8 | 2 |
| Half-Cheetah | 500 | 10, 100 | 200, 500, 700 | 0.995, 0.999 | 17 | 6 |

Table 1: Parameters of the environments.

| Environment | $\sigma$ | $\eta$ | Type |
|---|---|---|---|
| Dam control | 0.5 | 0.005 | Adam |
| Swimmer | 1 | 0.001 | Constant |
| Half-Cheetah | 0.1 | 0.001 | Adam |

Table 2: Training Parameters for the environments.

### B.2 Additional Results

In the following, we present the entire experimental campaign not present in the main paper.

**Swimmer.** The additional results presented on the Swimmer environment demonstrate the impact of the parameter $W$ w.r.t. longer horizons, higher discount factors and batch size. As shown in Figures 5, 6 and 7, for $H \in \{200, 500, 700\}$ independently from the batch size, we can appreciate how higher values of $W$ bring an improvement in terms of sample complexity converging faster to the optimal behavior, especially with higher values of $\gamma$. This result supports the statements of Theorem 4.4, giving a practical example of the benefits of a reduced effective horizon. Moreover, we can appreciate how searching in the space of $W$ policies induces a different manifold which is shown to help prevent plateau on the optimization phase (Figures 6 and 7).

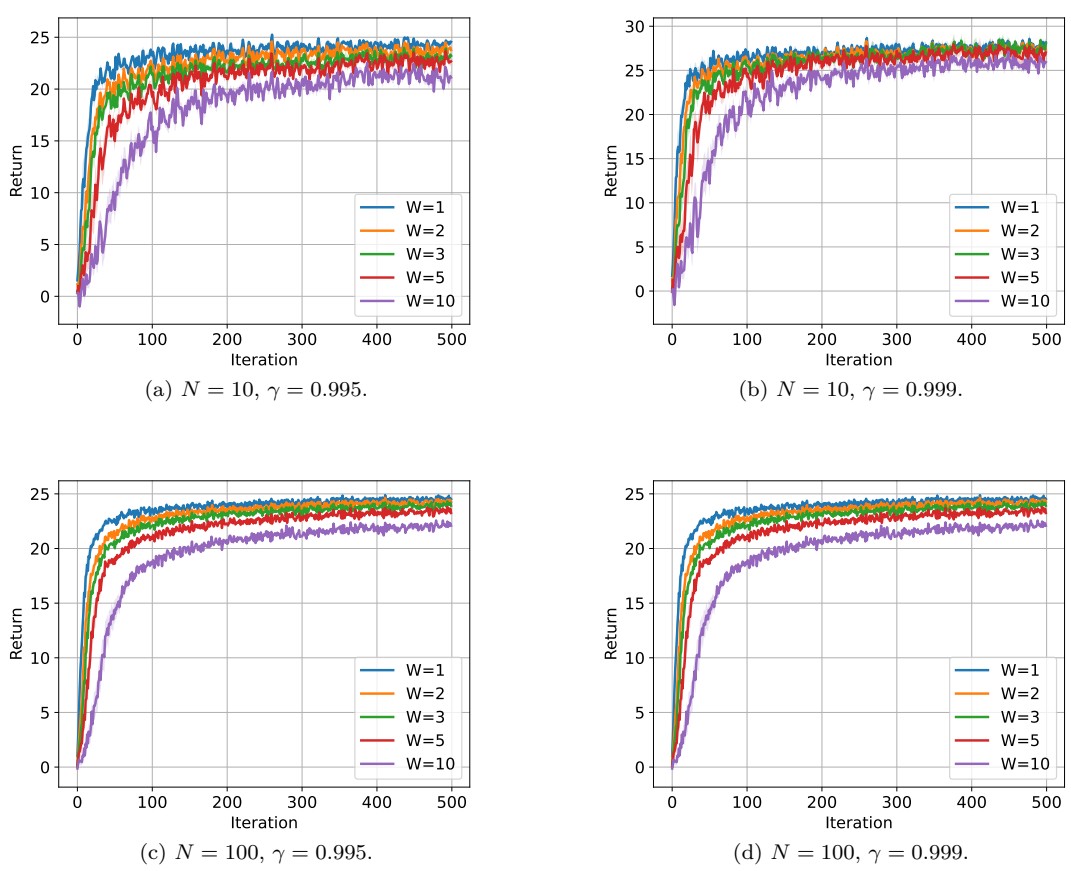

(a) $N = 10$, $\gamma = 0.995$.

(b) $N = 10$, $\gamma = 0.999$.

(c) $N = 100$, $\gamma = 0.995$.

(d) $N = 100$, $\gamma = 0.999$.

Figure 5: Learning curves for Swimmer with $H = 200$ (10 runs, mean $\pm$ 95% C.I.).

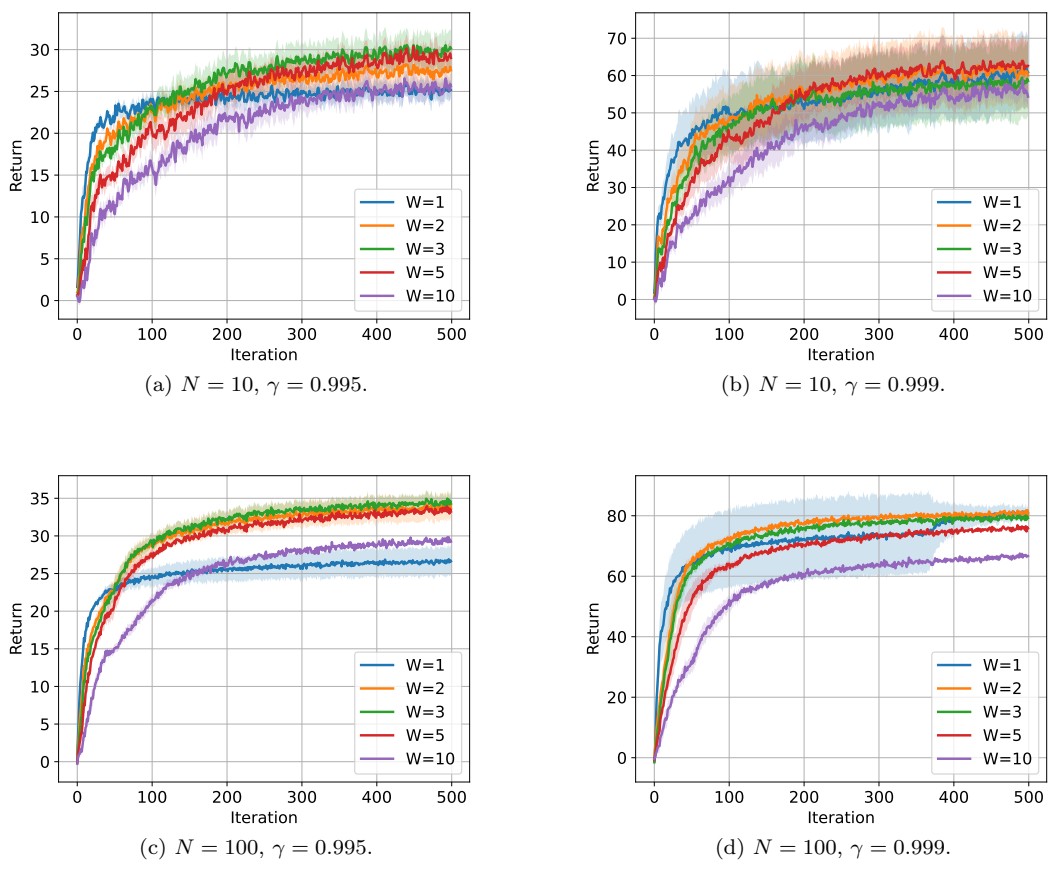

(a) $N = 10$, $\gamma = 0.995$.

(b) $N = 10$, $\gamma = 0.999$.

(c) $N = 100$, $\gamma = 0.995$.

(d) $N = 100$, $\gamma = 0.999$.

Figure 6: Learning curves for Swimmer with $H = 500$ (10 runs, mean $\pm$ 95% C.I.).

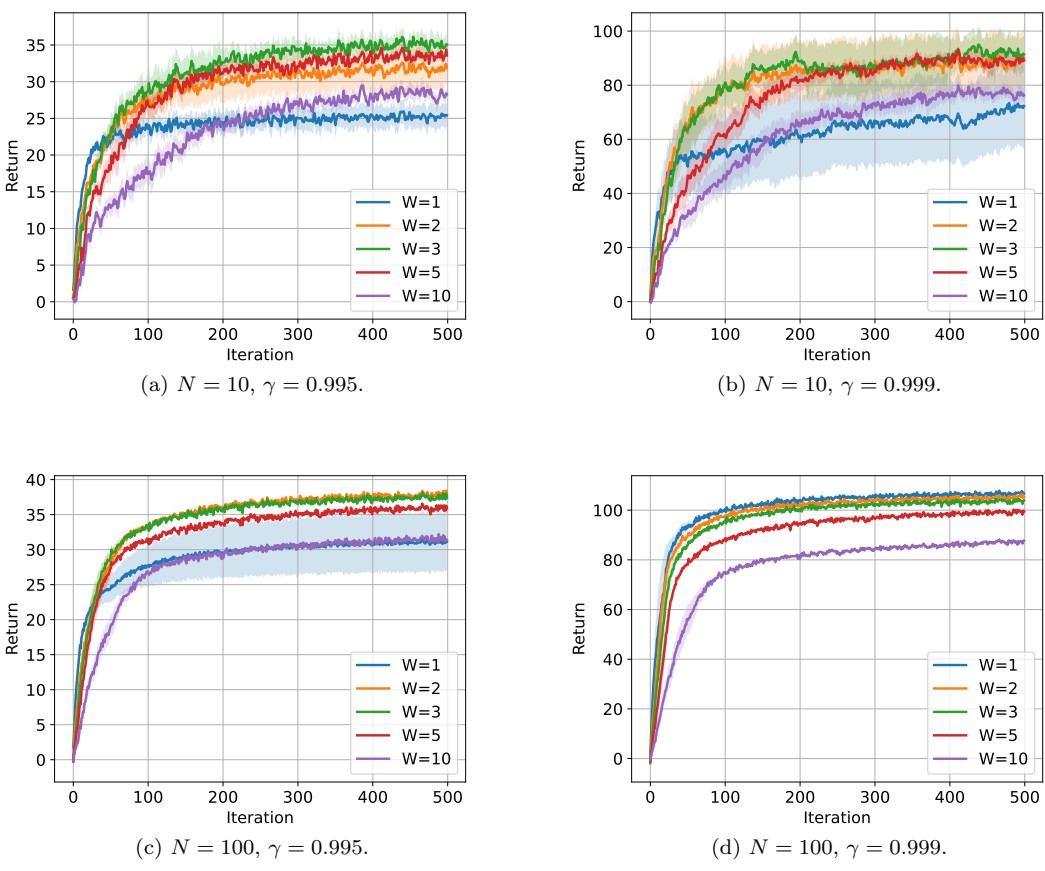

(a) $N = 10$, $\gamma = 0.995$.

(b) $N = 10$, $\gamma = 0.999$.

(c) $N = 100$, $\gamma = 0.995$.

(d) $N = 100$, $\gamma = 0.999$.

Figure 7: Learning curves for Swimmer with $H = 700$ (10 runs, mean $\pm$ 95% C.I.).

**Half-Cheetah.** The additional results presented on the Half-Cheetah environment demonstrate the impact of the parameter $W$ w.r.t. longer horizons, higher discount factors and batch size. We mostly appreciate how with small a small batch size ($N = 10$) we are able to estimate less noisy gradient, thus providing more accurate policy updates. We see in Figure 8, 9 and 10, how, regardless of the discount factor, values of $W > 1$ manage to obtain comparable performance to the closed-loop approach. Moreover, when the discount factor is higher ($\gamma = 0.999$) higher values of $W$ manages also to converge faster w.r.t. $W = 1$, with the benefits of a reduced computational complexity (Figure 4b). These results prove practically what is stated in Lemma 4.1, showing how the adoption of $W$-MDPs can effectively reduce the variance of the gradient estimator. Moreover, in Figures 9 and 10, we can observe how for longer horizons and high discount factors, values of $W > 1$ not only converge faster w.r.t the closed-loop approach, but in certain setting also outperform the $W = 1$ solution, providing an empirical demonstration of the benefits of a reduced effective horizon.

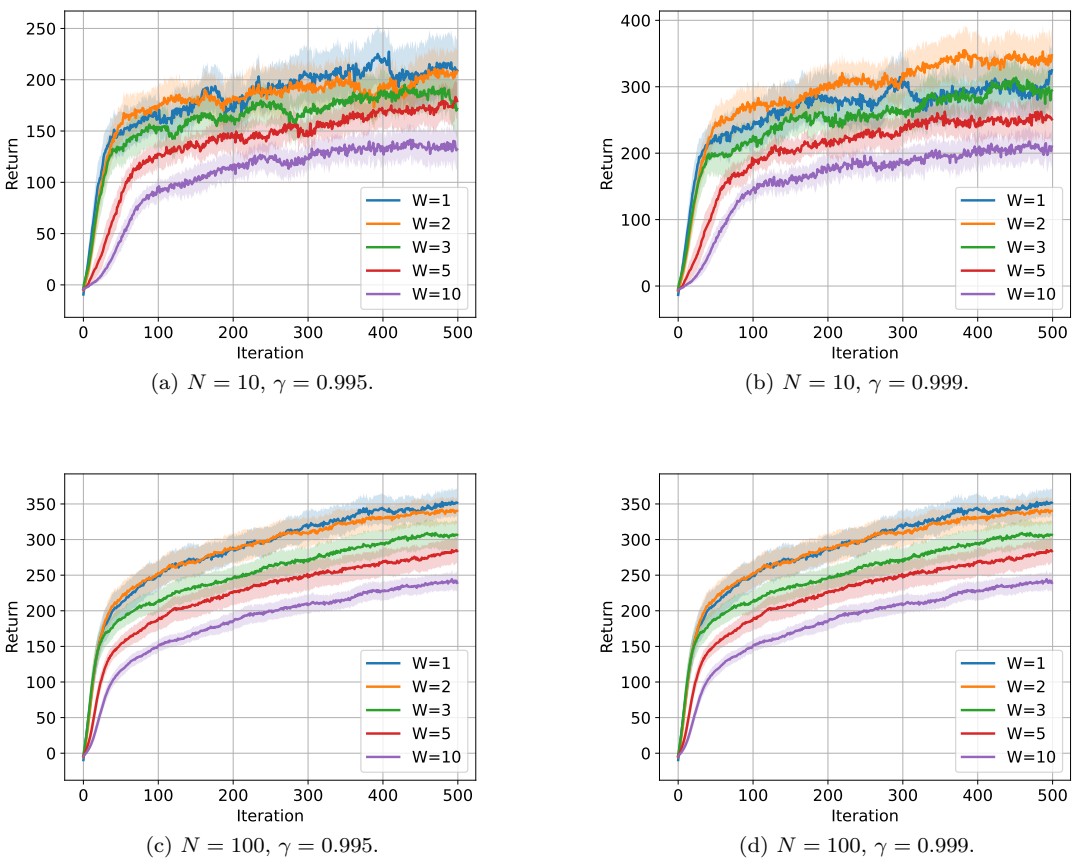

(a) $N = 10$, $\gamma = 0.995$.

(b) $N = 10$, $\gamma = 0.999$.

(c) $N = 100$, $\gamma = 0.995$.

(d) $N = 100$, $\gamma = 0.999$.

Figure 8: Learning curves for Half-Cheetah with $H = 200$ (10 runs, mean $\pm$ 95% C.I.).

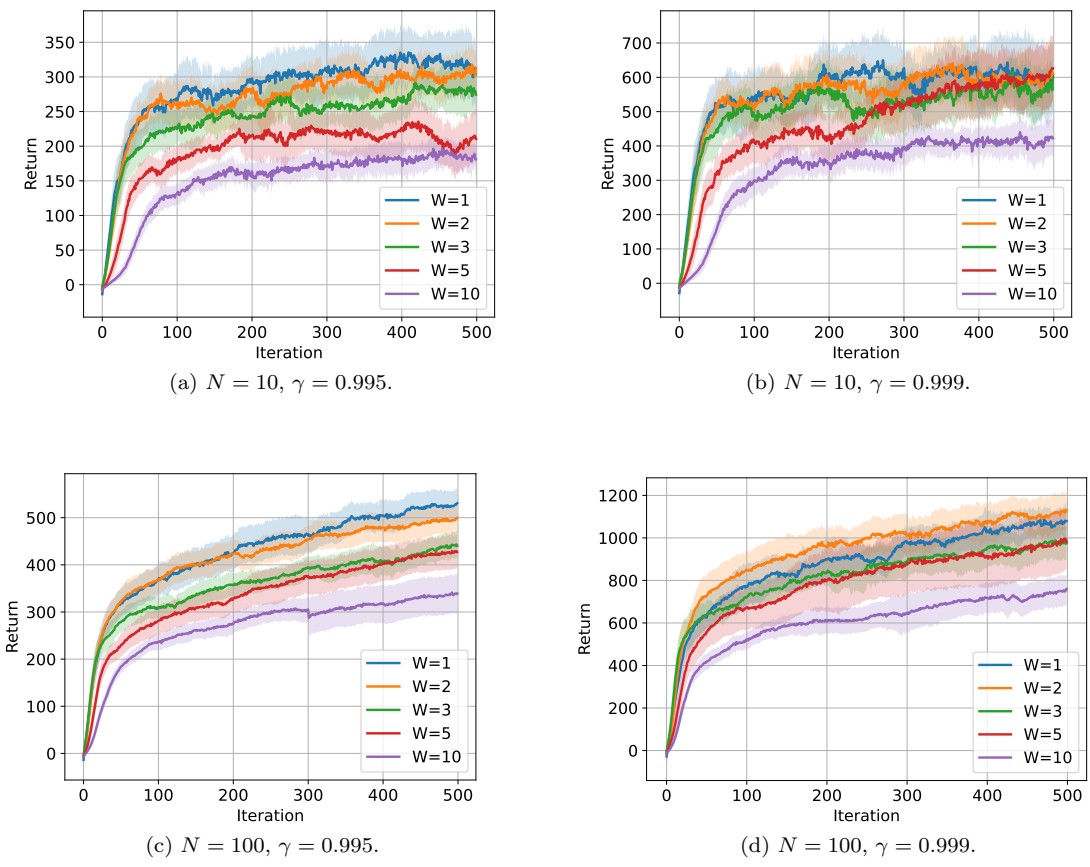

(a) $N = 10$, $\gamma = 0.995$.

(b) $N = 10$, $\gamma = 0.999$.

(c) $N = 100$, $\gamma = 0.995$.

(d) $N = 100$, $\gamma = 0.999$.

Figure 9: Learning curves for Half-Cheetah with $H = 500$ (10 runs, mean $\pm$ 95% C.I.).

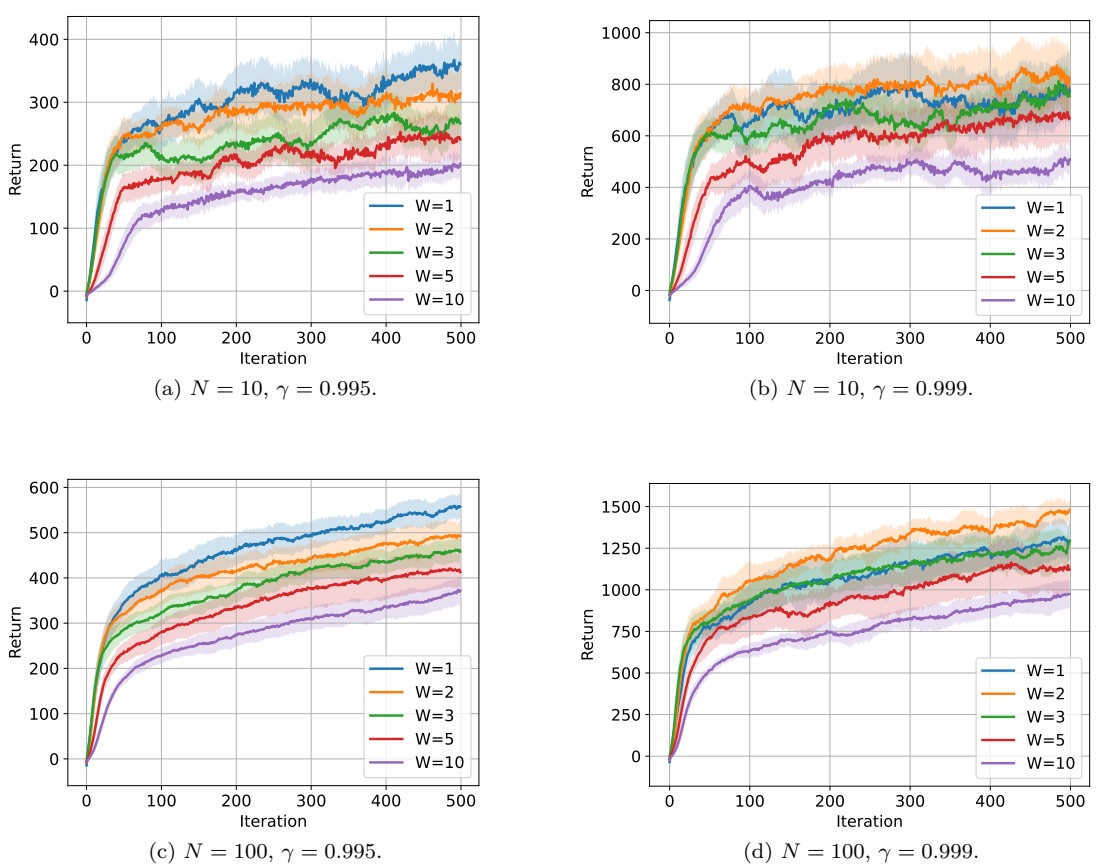

Figure 10: Learning curves for Half-Cheetah with $H = 700$ (10 runs, mean $\pm$ 95% C.I.).

