# OpenReview forum: "Trading-off Statistical and Computational Efficiency via $W$-step MDPs: A Policy Gradient Approach"
_TMLR — Rejected by TMLR_

### Review · Reviewer_rx4n · 2025-08-12

**Summary Of Contributions:**

The authors provide the framework of W-MDP, which generalizes the notion of standard MDP. In W-MDP, W actions are taken sequentially without observing the intermediate states and reward values. An aggregate W-horizon reward is supplied at the end of Wth action. The authors claim that by choosing an appropriate value of W, they can obtain a perfect balance between computational complexity and sample complexity.

**Audience:**

Yes

**Audience Explanation:**

The idea of aggregating actions to reduce training time may be useful to train policies in various contexts. Some of the TMLR audiences may be interested in the results.

**Claims And Evidence:**

Yes

**Claims Explanation:**

The authors have provided empirical evidence and theoretical insights to support their claims.

**Requested Changes:**

1. Lower bounds of the local and global sample complexities are provided in Theorems 4.3 and 4.4. Can the authors provide upper bounds on the sample complexities? Without an upper bound, the sample complexities could theoretically be arbitrarily large.

2. It seems that Theorems 4.3 and 4.4 are slightly modified versions of the results presented in (Papini 2020) and (Montenegro et al., 2024), respectively, and not novel contributions of the paper. This should be highlighted in the main text.

3. The lower bound on global sample complexity presented in the paper is $\tilde{\mathcal{\Omega}}(\epsilon^{-3})$. However, the state-of-the-art global sample complexity result is $\tilde{\mathcal{O}}(\epsilon^{-2})$ [1]. This shows that the lower bound presented in Theorem 4.4 is wrong/misspelt. Please compare Theorem 4.4 with the results given in [1].

4. I do not agree with the idea that theoreticians focus on sample complexity while practitioners study computational complexity. Sample complexity takes centre stage in many applications where the simulator is either unavailable or unreliable, and therefore, the policies need to be trained on a real environment. Moreover, even if a reliable simulator is available, simulating a single transition might be computationally extensive, especially for complex environments. In this case, reducing the sample complexity becomes the priority.

5. The authors have mentioned that (page 6) increasing W reduces the queries to the policy network. I do not understand why. It seems to me that the number of actions generated should remain the same. Only some computational load might get reduced because intermediate states are not observed.

6. In many applications, querying a policy network to generate actions may not be computationally expensive. However, as mentioned earlier, simulating a complex environment might be a demanding task. Therefore, beyond simple environment/control tasks, I am not entirely confident about the applicability/usefulness of this method.

[1] "Improved sample complexity analysis of natural policy gradient algorithm with general parameterization for infinite horizon discounted reward Markov decision processes." International Conference on Artificial Intelligence and Statistics. PMLR, 2024.

---

> ### Author Response · Authors · 2025-08-18
>
> We thank the Reviewer for the time spent reviewing our work and for the detailed feedback. Below, our answers to the Reviewer's questions and concerns. The revisions made in the manuscript according to the comments of the Reviewer has been marked in blue.
>
> > Lower bounds of the local and global sample complexities are provided in Theorems 4.3 and 4.4. Can the authors provide upper bounds on the sample complexities? Without an upper bound, the sample complexities could theoretically be arbitrarily large.
>
> > The lower bound on global sample complexity presented in the paper is $\tilde{\Omega}(\epsilon^{-3})$. However, the state-of-the-art global sample complexity result is $\tilde{\mathcal{O}}(\epsilon^{-2})$ [1]. This shows that the lower bound presented in Theorem 4.4 is wrong/misspelt. Please compare Theorem 4.4 with the results given in [1].
>
> We thank the Reviewer for pointing this out. In fact, Theorems 4.3 and 4.4 were misspelt already provide upper bounds on the local and global sample complexities, corresponding to the gradient update rule and the overall algorithm, respectively. As a consequence, being an upper bounds, it does not contradict the results in the literature. We have revised the notation in both theorems to make this clearer in the text.
>
> > It seems that Theorems 4.3 and 4.4 are slightly modified versions of the results presented in (Papini 2020) and (Montenegro et al., 2024), respectively, and not novel contributions of the paper. This should be highlighted in the main text.
>
> We agree with the Reviewer that the results of this paper make use of results derived in previous works from (Papini, 2020) and (Montenegro et al., 2024). Indeed, in the proofs of the statements (see Appendix A), we recognized their contributions. We added a reference to their works also in the main text.
>
> > I do not agree with the idea that theoreticians focus on sample complexity while practitioners study computational complexity. Sample complexity takes centre stage in many applications where the simulator is either unavailable or unreliable, and therefore, the policies need to be trained on a real environment. Moreover, even if a reliable simulator is available, simulating a single transition might be computationally extensive, especially for complex environments. In this case, reducing the sample complexity becomes the priority.
>
> We are sorry for the misunderstanding. Our point was that while theoreticians often disregard computational complexity, practitioners consider it in addition to sample complexity. Thus, in this paper, we adopt the latter perspective, analyzing such a trade-off while also providing guarantees on sample complexity. We have fixed this discussion in the manuscript to better clarify this point.
>
> > The authors have mentioned that (page 6) increasing W reduces the queries to the policy network. I do not understand why. It seems to me that the number of actions generated should remain the same. Only some computational load might get reduced because intermediate states are not observed.
>
> We thank the Reviewer for the question. Consider an action which belongs to $\mathbb{R}^d$. In a standard MDP setting, during a single episode of length $H$, the agent observes the state $H$ times and consequently queries the policy network $H$ times, each query producing an action in $\mathbb{R}^d$. In contrast, under the $W$-MDP formulation, the agent queries the policy network only $H/W$ times, and each query produces an action in $\mathbb{R}^{dW}$ (the concatenation of $H$ actions in $\mathbb{R}^d$). This leads to fewer policy network evaluations while preserving the same overall number of actions played. As shown in Figures 2b, 3b, and 4b and further discussed in the experimental section, this reduction in queries is advantageous in practice.
>
> > In many applications, querying a policy network to generate actions may not be computationally expensive. However, as mentioned earlier, simulating a complex environment might be a demanding task. Therefore, beyond simple environment/control tasks, I am not entirely confident about the applicability/usefulness of this method.
>
> We agree with the Reviewer that the environment is often computationally demanding and can significantly contribute to the overall training cost. Our work focuses on reducing the part of the computational burden that we can directly control, namely the algorithm itself. At the same time, we acknowledge that lowering the total computational cost also requires addressing the overhead introduced by the environment. For this reason, we studied the sample complexity of our method to ensure that the number of interactions with the environment remains as low as possible. In this way, our effort to reduce the computational cost does not come at the expense of higher sample complexity (that in turn will increase again the overall computational load).

---

### Review · Reviewer_Jqzd · 2025-08-24

**Summary Of Contributions:**

This paper proposes a W-step MDP framework to enhance efficiency from both theoretical and practical perspectives, aiming to reduce sample complexity and training time. In this framework, given a current state, the policy executes a sequence of actions and then collects the discounted W-step cumulative reward as the aggregated reward, transitioning to the resulting state after this sequence. The paper first provides a theoretical analysis of suboptimality under stochasticity, followed by an examination of variance reduction and sample efficiency for policy gradient methods as a specific instance. Finally, the authors present comprehensive experiments on three benchmarks, including detailed ablation studies, to validate the theoretical insights and demonstrate computational efficiency.

### Strengths
- The paper introduces a clear and well-motivated core idea.
- It combines both theoretical analysis and empirical validation, making the contribution well-rounded.
- The proposed method is conceptually simple and easily adaptable to standard reinforcement learning algorithms in practice.
- The W-step MDP concept aligns well with practical scenarios, such as robotics applications.

### Weaknesses
- Clarity in Experimental Section: initially, the connection between theoretical results (lemmas/theorems) and experiments is not immediately evident, as different theoretical components are supported by different benchmarks. While additional ablation studies in the appendix clarify this, it would strengthen the paper if the main experimental section explicitly indicated that these conclusions are supported by detailed ablations in the appendix.
- Self-Contained Theoretical Analysis: the theory section heavily relies on referencing external lemmas and theorems without providing derivations or sufficient intermediate steps. Since the theoretical component is not excessively long, it would be better to include the necessary derivations for completeness, rather than presenting results as a direct aggregation of prior assumptions and lemmas.
- Missing Definition: please define $R_{max​}$ explicitly for clarity.
- Minor Typographical Issue: in Theorem 2.1, add a space between “let” and $M_W​$. Ensure the same correction is applied in the appendix.

**Audience:**

Yes

**Audience Explanation:**

From an application standpoint, many engineering systems can only execute a sequence of actions before receiving feedback due to hardware constraints or practical considerations. This work provides valuable theoretical insights and problem formulation guidance that can benefit practitioners working in such settings.

**Claims And Evidence:**

Yes

**Claims Explanation:**

I think the authors’ claims are mostly reasonable. However, I have some questions regarding the choice of W and the reason why W > 1 (e.g., W=2) gets better return than W=1. Both points should be made much clearer to support the main idea.
- Choice of W: While the W-step concept provides flexibility, we only know that W=1 may not be optimal.
How do you select an appropriate W to trade off bias and variance in an unknown environment with unknown stochasticity?
- Performance of W > 1: Can you explain more why W > 1 (e.g., W=2) gets better return than W=1? You mention that a different manifold might avoid plateaus during optimization—can you elaborate on this reasoning? Are there any papers supporting this explanation?

**Requested Changes:**

- Is the “computational performance” mentioned in the abstract referring to computational efficiency?
- Experimental details
   - How many trials/random seeds were run for each curve?
   - Does the shaded area for the curves reflect the variance reduction you mention by reducing the effective horizon? If yes, can you explain why W=10 (purple curve in Figure 2) has the largest shaded area?
- Trade-off between statistical and computational efficiency
   - In the title, you mention Trading-off Statistical and Computational Efficiency. I would like to clarify: does your conclusion indicate there is a trade-off between statistical and computational efficiency? Or does your proposed method actually improve both statistical and computational efficiency simultaneously?
   - If it is a trade-off, I do not think it is appropriate to visualize subfigures a and c separately. A better approach might be: Identify the stop point for each W-step. Once it converges, report the running time and number of iterations (samples).
Use a Pareto Front / Trade-off Curve, with x-axis as runtime, y-axis as iterations, and the color of curves or dots indicating return values.
- Algorithm clarification: What is the subscript of s in $r(s, a_W) in Algorithm 3?
- Policy evaluation as a bottleneck: You mention:
> While the single queries may be more costly due to the extended temporal abstraction of the action, the overall reduction in query frequency leads to an improvement in the training phase, especially in settings where the policy evaluation might be a bottleneck.
   - Can you provide examples or scenarios where policy evaluation is indeed a bottleneck?

---

> ### Author Response · Authors · 2025-09-05
>
> We thank the Reviewer for the time spent reviewing our work and for the detailed feedback. Below, our answers to the Reviewer's questions and concerns. The revisions made in the manuscript according to the comments of the Reviewer have been marked in purple. Questions are synthesized due to space constraints.
>
> ## Weaknesses
>
> > Clarity in Experimental Section [...]
>
> We thank the Reviewer for the suggestion. We added a reference to the appendix as suggested in the experimental section of the main paper.
>
> > Self-Contained Theoretical Analysis: [...]
>
> We thank the Reviewer for the suggestion. We added in Appendix A1 all the assumptions and results needed to derive our results, making the manuscript more self-contained.
>
> > Missing Definition: please define $R_{max}$ explicitly for clarity.
>
> We added a note on $R_{\text{max}}$ in Section 2.1 specifying that it is a scalar positive value.
>
> > Minor Typographical Issue: in Theorem 2.1, add a space between “let” and $M_W$. Ensure the same correction is applied in the appendix.
>
> We fixed it, thanks.
>
> ## Claims
>
> > Choice of $W$: While the $W$-step concept provides flexibility, we only know that $W=1$ may not be optimal. How do you select an appropriate W [...]?
>
> We thank the Reviewer for the interesting question. In fact, if we don't know the level of stochasticity of the task, as a general rule, it is better to maintain a low value of $W$. This remains an interesting open question, which we added in the conclusion section as a future work to be investigated.
>
> > Performance of $W > 1$ [...]
>
> We thank the Reviewer for the question. With a value of $W$ other than $1$, the optimization space changes, so we could have better results due to the use of a different policy class that will induce a different manifold. Despite this, our conjecture is not supported by theoretical guarantees. We specified this is a conjecture also in the paper.
>
> ## Requested changes
>
> > Is the "computational performance" mentioned in the abstract referring to computational efficiency?
>
> Yes, with "computational performance" we refer to computational efficiency. We changed it for coherence with the other parts.
>
> > Experimental details. How many trials/random seeds [...]?
>
> We thank the Reviewer for the helpful comment. This was an oversight on our side. We have now added, below each figure, the number of trials and the meaning of the shaded areas.
>
> > Does the shaded area for the curves reflect the variance reduction you mention by reducing the effective horizon? [...]
>
> No, with the shaded areas in the learning curves, we intend the variance of the results between the various trials, while in the analysis, we evaluate the variance of the gradient estimator. Clearly, the two quantities are related, but not directly proportional.
>
> > Trade-off between statistical and computational efficiency. [...]
>
> We, indeed, concluded that between statistical and computational performance, there is a trade-off. However, in our work, we show how it is possible to maintain competitive statistical complexity, as presented in Theorems 4.3/4.4 (from a theoretical standpoint) and in Section 5 (from an empirical perspective), while also improving computational efficiency, as shown in Section 5.
>
> > If it is a trade-off, I do not think it is appropriate to visualize subfigures a and c separately. [...]
>
> We decided to use this type of graphic presentation to demonstrate that, although from a statistical point of view (i.e., in terms of iterations/samples), the curve with $W = 1$ accelerates more rapidly, from a computational point of view (i.e., in terms of execution time), the same configuration is the slowest among those studied.
>
> > Algorithm clarification: What is the subscript of s in $r(s, a_W)$ in Algorithm 3?
>
> We thank the Reviewer for pointing out this typo. Indeed, the $s$ presented in the reward computation inside the $W$-loop need $h + w$ as a subscript as the reward is computed by starting in the observed state at time $h$ plus the offset of the duration of the sequence until now $w$ (as defined in Section 2.2). We fixed this typo in the manuscript.
>
> > Policy evaluation as a bottleneck [...]
>
> We thank the Reviewer for the question. Policy evaluation can become a bottleneck, particularly in scenarios when the evaluation/querying of the policy involves running complex simulations, high-dimensional environments, or costly rollouts (e.g., physics-based control tasks, robotics simulators). Our contribution focuses on reducing the computational cost of the learning algorithm, ensuring that the frequency of the policy evaluation is kept low. At the same time, we recognize that the overall training cost also depends on the environment overhead. For this reason, we tried to balance the number of interactions with the environment to keep them as low as possible, while maintaining satisfactory learning performance.

---

### Review · Reviewer_zz9R · 2025-08-25

**Summary Of Contributions:**

This paper introduces $W$-MDPs, which, given an MDP, extend the action space to be composed of a sequence of $W$ actions of the original MDP, and accumulate the rewards taking into account the discount factors. In this case, the agent only acts and observes the state every $W$ time steps. The contributions include: (1) a theoretical bound on the performance loss when using $W$-MDPs compared to standard MDPs, highlighting the role of environment stochasticity; (2) a method named MetaStep that adapts existing RL algorithms to $W$-MDPs; and (3) integration with the GPOMDP policy gradient method, providing theoretical analysis of variance reduction and convergence improvements. Experiments are conducted in a Dam control environment, and the Swimmer and Half-Cheetah robots from the MuJoCo benchmark.

Strengths:
- Temporal abstractions are a very relevant topic in the RL literature, which draws much interest from the community.
- The theoretical analysis of the variance, sample complexity, and optimality gap of $W$-MDPs is sound and may be potentially useful for researchers working on similar topics.

 Weaknesses:
- No baselines were considered in the experiments from the “action persistence” class of methods, or from the hierarchical RL literature. There is also a limited literature review of similar approaches.
- The novelty of the method is limited, since very similar approaches have been widely studied, and these approaches have not been empirically compared in the experimental section. For instance, the Option Keyboard loop (Barreto et al., 2019) is very similar in essence to Algorithm 1.
- The paper is missing a discussion on some of the limitations of the method, especially regarding the increased action space.
- It is disappointing that the best value for $W$ found in the experiments is 2, suggesting limited gains from the proposed approach.
- A few claims of the paper are currently not supported by clear evidence (see below).

**Additional Comments:**

Minor typo: “than” $D(P) = 0$ -> “then”.

**Audience:**

No

**Audience Explanation:**

Temporal abstractions are, in general, a very relevant topic in the RL literature, which draws much interest from the community. I think the theoretical analysis introduced in this paper might be useful for some researchers working on similar topics. However, given the limited novelty of the method and the limited performance gains found in the experiments, the interest in the paper may not be very significant.

**Broader Impact Concerns:**

NA.

**Claims And Evidence:**

No

**Claims Explanation:**

Two claims (although not the main claims of the paper) are not currently supported by evidence:

- In Fig. 2(a), the authors state that “Notably, with W = 2, convergence occurs slightly faster than with the single-step approach, confirming the sample efficiency gains of a reduced effective horizon”. I do not think this is clearer from this figure.

- In Fig. 3, the authors claim that the faster convergence “is due to the search in the space of $\Pi^W$ policies, which induces a different manifold that might avoid plateaus during optimization”. What is the evidence for this claim? It may be a good hypothesis, but this is not shown in the paper.

**Requested Changes:**

- No baselines were considered in the experiments from the “action persistence” class of methods, or from the hierarchical RL literature. There is also a limited literature review of similar approaches. I suggest including one or more of the baselines discussed in Section 6 (e.g., the works cited in paragraph “Action Persistance”).

- Regarding GPOMDP, the authors state that it is “a well-known policy gradient method” and “a widely-adopted action-based policy gradient algorithm”. Although I agree that (Baxter and Barlett, 2001) is an influential work in the 2000s, I am not sure it is widely adopted now. Can the authors point to recent works that adopt this algorithm and discuss its current relevance?
- Why did the authors not consider state-of-the-art RL algorithms such as PPO, SAC, DQN, etc?

- The authors should discuss the factorial number of action permutations that result in a significant increase in the size of the action space in Section 2.2 - Definition 2.1.

- In “We sample a sequence of $W$ actions from the parametric $\pi^{W}_{\theta}$”, it is not clear whether the policy is queried exactly $W$ times, or whether the output of the policy is a vector/tuple of dimension $W$. If it is the latter, then the authors should discuss the fact that the action space is increasing significantly. It would also be useful to make it clearer if the method is assuming discrete or continuous action spaces before Section 5, when it is clearer that the authors are considering continuous control tasks.

- The authors should state how many random seeds were used in the experiments, and what the shaded region in the plots is (e.g., confidence interval). How were the hyperparameters chosen?

- In Fig. 3(a), why can we only see a shaded region for the blue curve? Is the variance equal to 0 in all other curves? Please clarify.

- It is a bit disappointing that the best value for $W$ is 2. I suggest that the authors discuss why the method shows limited empirical gains with higher values for $W$.

- “In our work, we propose a task-agnostic solution that embraces open-loop control methodologies but provides the agent with feedback after a fixed number of decision-making steps.” Although this is true, this has been done in the literature of hierarchical RL multiple times. See [1]-[5] below.

- In the discussion of Action Persistence, the authors state that “However, these approaches tend to be too rigid and require properly setting the task control frequency. Furthermore, they poorly scale with the increase in dimensionality of the action space. Our work aligns with the philosophy of action persistence but provides more flexibility to the framework.”
Please clarify what you mean by “task control frequency”. Note that the proposed method also requires setting a good value for $W$. Moreover, the proposed method also does not scale well with the increase in dimensionality of the action space (e.g., the best performance is achieved with $W=2$).

1. Sutton, Precup, & Singh (1999) – Options framework
 Sutton, R. S., Precup, D., & Singh, S. (1999). Between MDPs and semi-MDPs: A framework for temporal abstraction in reinforcement learning. Artificial Intelligence, 112(1–2), 181–211. https://doi.org/10.1016/S0004-3702(99)00052-1
2. Parr & Russell (1998) – Hierarchies of Abstract Machines (HAMs)
 Parr, R., & Russell, S. (1998). Reinforcement learning with hierarchies of machines. In Proceedings of the 1997 Conference on Advances in Neural Information Processing Systems (pp. 1043–1049). MIT Press.
3. Dietterich (2000) – MAXQ decomposition
 Dietterich, T. G. (2000). Hierarchical reinforcement learning with the MAXQ value function decomposition. Journal of Artificial Intelligence Research, 13, 227–303. https://doi.org/10.1613/jair.639
4. Bacon, Harb, & Precup (2017) – Option-Critic Architecture
 Bacon, P.-L., Harb, J., & Precup, D. (2017). The option-critic architecture. In Proceedings of the Thirty-First AAAI Conference on Artificial Intelligence.
5. Barreto et al. (2019) “Option Keyboard” paper
Barreto, A., Borsa, D., Hou, S., Comanici, G., Aygün, E., Hamel, P., Toyama, D., Hunt, J., Mourad, S., Silver, D., & Precup, D. (2019). The Option Keyboard: Combining Skills in Reinforcement Learning. In Advances in Neural Information Processing Systems 32.

---

> ### Author Response · Authors · 2025-09-05
>
> We thank the Reviewer for the time spent reviewing our work and for the detailed feedback. Below, our answers to the Reviewer's questions and concerns. The revision made in the manuscript according to the comments of the Reviewer has been marked in orange.
>
> ## Weaknesses
>
> > No baselines were considered in the experiments from the “action persistence” class of methods, or from the hierarchical RL literature. There is also a limited literature review of similar approaches.
>
> We thank the Reviewer for pointing out this aspect in our work. The philosophy of the project is to take a single gradient-based optimization methodology and compare its closed-loop version with the different $W$-step variants. We do not aim to get state-of-the-art performance within this work, but to take an algorithm that is theoretically analyzable and compare the two variants. On the other hand, we agree with the reviewer that the literature review in the HRL field is limited, so we worked to integrate the related works on this side.
>
> > The novelty of the method is limited, since very similar approaches have been widely studied, and these approaches have not been empirically compared in the experimental section. For instance, the Option Keyboard loop (Barreto et al., 2019) is very similar in essence to Algorithm 1.
>
> We thank the Reviewer for bringing to our attention the Option Keyboard (OK) loop. However, even if our work and the OK framework share some general concepts at their core, we do not think they lie on the same ground. First, the OK is built upon the Option Framework (Precup, 1999), which differs from ours. Moreover, the algorithm needs to learn first the base options, computing all pairwise value functions. In our work, we do not need this step as we adopt the same policy for the entire task without the necessity to define options. However, the work is strongly related to ours, and we added it to the related works section. Thank you for pointing it out.
>
> > The paper is missing a discussion on some of the limitations of the method, especially regarding the increased action space.
>
> We agree with the Reviewer on the necessity of a discussion, limitation-wise, regarding the increase in the action space. We added a discussion on that at the end of the experimental section and in the conclusions (see Limitations).
>
> > It is disappointing that the best value for $W$ found in the experiments is 2, suggesting limited gains from the proposed approach.
>
> We agree with the Reviewer that in the proposed settings, the optimal value of $W = 2$. However, we expect that in different tasks where the action played, or small variations of it, can be persisted for longer, the optimal $W$ will be higher. This conjecture is supported by the fact that our work generalizes the concept of action persistence, allowing us to play action sequences structured by different actions (maybe similar), w.r.t. a single repeated action.
>
> ## Claims
>
> > In Fig. 2(a), the authors state that “Notably, with W = 2, convergence occurs slightly faster than with the single-step approach, confirming the sample efficiency gains of a reduced effective horizon”. I do not think this is clearer from this figure.
>
> We agree with the Reviewer, and we removed the sentence.
>
> > In Fig. 3, the authors claim that the faster convergence “is due to the search in the space of $\Pi^W$ policies, which induces a different manifold that might avoid plateaus during optimization”. What is the evidence for this claim? It may be a good hypothesis, but this is not shown in the paper.
>
> We thank the reviewer for pointing this out. The hypothesis stated on the ablation of Fig. 3 is a conjecture based on empirical evidence. We adjusted the claim made in the manuscript so that the reader would better understand that this is a conjecture, not based on theoretical evidence.
>
> ## Interest
>
> > Temporal abstractions are, in general, a very relevant topic in the RL literature, which draws much interest from the community. I think the theoretical analysis introduced in this paper might be useful for some researchers working on similar topics. However, given the limited novelty of the method and the limited performance gains found in the experiments, the interest in the paper may not be very significant.
>
> In our opinion, this should be intended as a first approach to this hybrid open-closed loop world. Our objective is to present the $W$-MDP framework and characterize it also in terms of performance loss and show the potential gain of this idea, not reaching the best empirical performance possible. Indeed, we choose to consider GPOMDP as it allows us to have both a theoretical-empirical discussion of the gain, even if in terms of absolute performance, other actor-critic algorithms (which can be adapted to this framework) we expect may perform better.

---

> > ### Author Response · Authors · 2025-09-05
> >
> > > No baselines were considered in the experiments from the “action persistence” class of methods, or from the hierarchical RL literature. There is also a limited literature review of similar approaches. I suggest including one or more of the baselines discussed in Section 6 (e.g., the works cited in paragraph “Action Persistence”).
> >
> > We thank the Reviewer for the comment. In this work, we did not aim to overperform state-of-the-art algorithms, but rather to take an analyzable algorithm (GPOMDP) and apply temporal abstraction techniques on top of it, studying its impact. However, our work is not intended to be just vanilla policy gradient, but the algorithm selected should just be seen as an example of a methodology that can be analyzed from both a theoretical and empirical point of view. The proposed work is a first step, which will be followed by projects that give greater importance to empirical results, thus relying on state-of-the-art approaches.
> >
> > ## Requested Changes
> >
> > > Regarding GPOMDP, the authors state that it is “a well-known policy gradient method” and “a widely-adopted action-based policy gradient algorithm”. Although I agree that Baxter and Barlett (2001) is an influential work in the 2000s, I am not sure it is widely adopted now. Can the authors point to recent works that adopt this algorithm and discuss its current relevance?
> >
> > The choice of GPOMDP is determined by the fact that the algorithm can also be studied from a theoretical point of view. One of our main claims regards the statistical complexity obtained by our algorithm, which would have been harder to derive using different types of algorithms for which there are still no studies of the same theoretical level (e.g., PPO, SAC, DQN).
> >
> > > Why did the authors not consider state-of-the-art RL algorithms such as PPO, SAC, DQN, etc?
> >
> > We thank the Reviewer for the question. In this work, we purposely did not consider those types of algorithms because the focus of the work is not to overperform state-of-the-art RL algorithms, but rather to propose a new methodology that can be paired with the latter. Specifically, in this work, we decide to adopt GPOMDP as it is analyzable theoretically and empirically. We added a note on the suggested algorithm in the future works.
> >
> > > The authors should discuss the factorial number of action permutations that result in a significant increase in the size of the action space in Section 2.2 - Definition 2.1.
> >
> > We thank the Reviewer for pointing this out. Indeed, especially in finite state action spaces, this phenomenon is more pronounced. In continuous space, the problem persists, albeit in a more mitigated form from an empirical perspective. Indeed, the fact that it is possible to use function approximators lightens this problem, as the actions that will be played are related to each other in practice. We added a comment on the action space, which increases, and we need to search for a trade-off between reduced effective horizon and increased action space at the end of the experimental section.
> >
> > > In “We sample a sequence of $W$ actions from the parametric $\pi_W^\theta$, it is not clear whether the policy is queried exactly $W$ times, or whether the output of the policy is a vector/tuple of dimension $W$. If it is the latter, then the authors should discuss the fact that the action space is increasing significantly.
> >
> > We confirm that the output is a vector of actions of dimension $W$. The action space indeed increases with the value of $W$. We added a discussion on this (see answer above).
> >
> > > It would also be useful to make it clearer if the method is assuming discrete or continuous action spaces before Section 5, when it is clearer that the authors are considering continuous control tasks.
> >
> > We added it in Section 2. Thank you for the suggestion.
> >
> > > The authors should state how many random seeds were used in the experiments, and what the shaded region in the plots is (e.g., confidence interval). How were the hyperparameters chosen?
> >
> > We thank the Reviewer for pointing this out; it was an oversight. We added the number of runs and the meaning of shaded areas in the captions. Regarding the hyperparameters, we chose them by taking inspiration from benchmarks for these environments.
> >
> > > In Fig. 3(a), why can we only see a shaded region for the blue curve? Is the variance equal to 0 in all other curves? Please clarify.
> >
> > As the reviewer rightly said, the shaded region on the other curves is barely visible because the results on the various runs have low variance. Specifically, in Fig. 3(a), this phenomenon is more noticeable since, as said in the main text, with $W = 1$ some of the trails end up in a plateau during the optimization process.

---

> > > ### Author Response · Authors · 2025-09-05
> > >
> > > > It is a bit disappointing that the best value for $W$ is 2. I suggest that the authors discuss why the method shows limited empirical gains with higher values for $W$.
> > >
> > > We thank the Reviewer for pointing this out. In our work, there is a major trade-off between lightening the curse of horizon and not increasing the action space too much. In the proposed setting, the optimal value of $W$ found is 2, but we do not exclude that in scenarios where the choice of actions does not require making decisions that are too dissimilar from each other, the best $W$ would be higher.
> > >
> > > > “In our work, we propose a task-agnostic solution that embraces open-loop control methodologies but provides the agent with feedback after a fixed number of decision-making steps.” Although this is true, this has been done in the literature of hierarchical RL multiple times. See [1]-[5] below.
> > >
> > > We thank the Reviewer for bringing this work to our attention. We agree that our work falls closely within the hierarchical RL field. However, there are some major differences between our work and the cited ones. First, we do not adopt a Semi-MDP framework. The usage of an action sequence can be seen as a more specific way of imaging an option, but with a more defined structure and a single shared policy. For what concerns works [3]-[5], they all share the Semi-MDP structure of the problem, thus needing some prior steps of option discovery/learning. In our work, we do not need to explicitly (or semantically) define an option as it is represented by an open-loop course of actions. We added the works suggested in the related works sections.
> > >
> > > > In the discussion of Action Persistence, the authors state that “However, these approaches tend to be too rigid and require properly setting the task control frequency. Furthermore, they poorly scale with the increase in dimensionality of the action space. Our work aligns with the philosophy of action persistence, but provides more flexibility to the framework.” Please clarify what you mean by “task control frequency”. Note that the proposed method also requires setting a good value for $W$. Moreover, the proposed method also does not scale well with the increase in dimensionality of the action space (e.g., the best performance is achieved with $W = 2$).
> > >
> > > We do agree with the Reviewer on the limitations of the choice of the optimal $W$ for a designed task, which, in the scope of this first work, is manually tuned. However, unlike the action persistence methodologies, we provide major flexibility in the choice of the action. When referring to setting "task control frequency," we mean that in action persistence methods, an action is repeated for a fixed number of time steps. This is equivalent to reducing the frequency at which we query the policy, perform a "choice-and-hold" of our action. Differently, our method allows the policy to output a sequence of $W$ actions directly, providing more flexibility, at the price of an increased action space. The actions within the sequence can vary rather than being identical, which can better adapt to the dynamics of the task.

---

### Decision · Action_Editor_WU5C · 2025-10-18

**Recommendation:** Reject

**Audience:**

Yes

**Audience Explanation:**

The proposed framework aligns with engineering implementations in applications. With enough theoretical and empirical evidence, the paper could be of great interest to the audience of TMLR.

**Claims And Evidence:**

No

**Claims Explanation:**

This paper introduces W-step MDPs, which extend a given MDP by expanding the action space to sequences of W actions from the original MDP, with rewards accumulated according to discount factors. In this formulation, the agent acts and observes the state only every W time steps. While the reviewers find that the idea of aggregating actions could be potentially useful in certain applications, they unanimously agree that the current evidence presented in the paper is insufficient.

Although the authors claim that W-step MDPs enhance state-of-the-art RL algorithms, the reported performance does not support this claim. The experimental comparison is too limited—only including GPOMDP—without considering recent and widely adopted policy gradient baselines such as PPO or SAC. Therefore, additional empirical evaluations are needed to provide convincing evidence for the claimed effectiveness of the proposed W-step MDPs and the MetaStep algorithm, particularly regarding their ability to improve computational efficiency while maintaining competitive sample efficiency.

**Resubmission Of Major Revision:**

The authors may consider submitting a major revision at a later time.